# Deep Graph Neural Networks with Shallow Subgraph Samplers

## Abstract

While Graph Neural Networks (GNNs) are powerful models for learning representations on graphs, most state-of-the-art models do not have significant accuracy gain beyond two to three layers. Deep GNNs fundamentally need to address: 1). *expressivity* challenge due to oversmoothing, and 2). *computation* challenge due to neighborhood explosion. We propose a simple "deep GNN, shallow sampler" design principle to improve both the GNN accuracy and efficiency — to generate representation of a target node, we use a deep GNN to pass messages only within a shallow, localized subgraph. A properly sampled subgraph may exclude irrelevant or even noisy nodes, and still preserve the critical neighbor features and graph structures. The deep GNN then smooths the informative local signals to enhance feature learning, rather than oversmoothing the global graph signals into just "white noise". We theoretically justify why the combination of deep GNNs with shallow samplers yields the best learning performance. We then propose various sampling algorithms and neural architecture extensions to achieve good empirical results. Experiments on five large graphs show that our models achieve significantly higher accuracy and efficiency, compared with state-of-the-art.

## 1 Introduction

Graph Neural Networks (GNNs) have now become the state-of-the-art models for graph mining (Wu et al., 2020; Hamilton et al., 2017b; Zhang et al., 2019), facilitating applications such as social recommendation (Monti et al., 2017; Ying et al., 2018; Pal et al., 2020), knowledge understanding (Schlichtkrull et al., 2018; Park et al., 2019; Zhang et al., 2020) and drug discovery (Stokes et al., 2020; Lo et al., 2018). With the numerous architectures proposed (Kipf & Welling, 2016; Hamilton et al., 2017a; Veličković et al., 2018), it still remains an open question how to effectively design *deep* GNNs. There are two fundamental obstacles that are intrinsic to the underlying graph structure:

- *Expressivity* challenge: deep GNNs tend to oversmooth (Li et al., 2018). They collapse embeddings of different nodes into a fixed low-dimensional subspace after repeated neighbor mixing.

- *Computation* challenge: deep GNNs recursively expand the adjacent nodes along message passing edges. The neighborhood size may grow exponentially with model depth (Chen et al., 2017).

Due to oversmoothing, one of the most popular GNN architectures, Graph Convolutional Network (GCN) (Kipf & Welling, 2016), has been theoretically proven as incapable of scaling to deep layers (Oono & Suzuki, 2020; Rong et al., 2020; Huang et al., 2020). Remedies to overcome the GCN limitations are two-folded. From the neural architecture perspective, researchers are actively seeking for more expressive neighbor aggregation operations (Veličković et al., 2018; Hamilton et al., 2017a; Xu et al., 2018a), or transferring design components (such as residual connection) from deep CNNs to GNNs (Xu et al., 2018b; Li et al., 2019; Huang et al., 2018). From the data perspective, various works (Klicpera et al., 2019a;b; Bojchevski et al., 2020) revisit classic graph analytic algorithms to reconstruct a graph with nicer topological property. The two kinds of works can also be combined to jointly improve the quality of message passing in deep GNNs.

All the above GNN variants take a "global" view on the input graph $\mathcal{G}(\mathcal{V}, \mathcal{E})$ — *i.e.*, all nodes are considered as belonging to the same $\mathcal{G}$, whose size can often be massive. To generate the node embedding, no matter how we modify the architecture and the graph structure, a deep enough GNN

would always propagate the influence from the entire node set $\mathcal{V}$ into a single target node. Intuitively, for a large graph, most nodes in $\mathcal{V}$ barely provide any useful information to the target nodes. We thus regard such "global view" on $\mathcal{G}$ as one of the root causes for both the expressivity and computation challenges discussed above. In this work, for the node embedding task, we take an alternative "local view" and interpret the GNN input as $\mathcal{V} = \bigcup_{v \in \mathcal{V}} \mathcal{V}_{[v]}$ and $\mathcal{E} = \bigcup_{v \in \mathcal{V}} \mathcal{E}_{[v]}$. In other words, each target node $v$ belongs to some small graph $\mathcal{G}_{[v]}$ capturing the characteristics of only the node $v$. The entire input graph $\mathcal{G}$ is observed as the union of all such local yet latent $\mathcal{G}_{[v]}$. Such simple global-to-local switch of perspective enables us to address both the expressivity and computation challenges *without* resorting to alternative GNN architectures or reconstructing the graph.

**Present work: SHADOW-GNN.** We propose a "Deep GNN, shallow sampler" design principle that helps improve the expressive power and inference efficiency of various GNN architectures. We break the conventional thinking that an $L$-layer (deep) GNN has to aggregate $L$-hop (faraway) neighbors. We argue that the GNN receptive field for a target node should be *shallower* than the GNN depth. In other words, an $L$-layer GNN should only operate on a small subgraph $\mathcal{G}_{[v]}$ surrounding the target node $v$, where $\mathcal{G}_{[v]}$ consists of (part of) the $L_0$-hop neighborhood. The deep *vs.* shallow comparison is reflected by setting $L_0 < L$. We name such a GNN on $\mathcal{G}_{[v]}$ as a SHADOW-GNN. We justify our design principle from two aspects. Firstly, *why do we need the neighborhood to be shallow*? As a motivating example, the average number of 4-hop neighbors for the `ogbn-products` graph (Hu et al., 2020) is 0.6M, corresponding to 25% of the full graph size. Blindly encoding the 0.6M node features into a single embedding vector can create the "information bottleneck" (Alon & Yahav, 2020). The irrelevant information from the majority of the 0.6M nodes may also "dilute" the truly useful signals from a small set of close neighbors. A simple solution to the above issues is to manually create a shallow neighborhood by subgraph sampling. The second question regarding SHADOW-GNN is: *why do we still need deep GNNs*? Using more number of layers than the number of hops means the same pair of nodes may exchange messages with each other multiple times. Intuitively, this helps the GNN better absorb the subgraph information. Theoretically, we prove that a GNN deeper than the hops of the subgraph can be more powerful than the 1-dimensional Weisfeiler-Lehman test (Shervashidze et al., 2011). A shallow GNN, on the contrary, cannot accurately learn certain simple functions such as unweighted mean of the shallow neighborhood features. Note that with GCN as the backbone, a SHADOW-GCN still performs signal smoothing in each layer. However, the important distinction is that a deep GCN smooths the full $\mathcal{G}$ regardless of the target node, while a SHADOW-GCN constructs a customized smoothing domain $\mathcal{G}_{[v]}$ for each target $v$. The variance in those smoothing domains created by SHADOW-GCN encourages variances in the node embedding vectors. With such intuition, our analysis shows that SHADOW-GNN does not oversmooth. Finally, since the sizes of the shallow neighborhoods are independent of the GNN depth, the computation challenge due to neighbor explosion is automatically addressed.

We propose various subgraph samplers for SHADOW-GNN, including the simplest $k$-hop sampler and a sampler based on personalized PageRank, to improve the inference accuracy and computation efficiency. By experiments on five standard benchmarks, our SHADOW-SAGE and SHADOW-GAT models achieve significant accuracy gains compared with the original GraphSAGE and GAT models. In the meantime, the inference cost is reduced by orders of magnitude.

## 2 RELATED WORK AND PRELIMINARIES

**Deep GNNs.** Recently, numerous GNN models (Kipf & Welling, 2016; Defferrard et al., 2016; Hamilton et al., 2017a; Veličković et al., 2018; Xu et al., 2018b;a) have been proposed. In general, the input to a GNN is the graph $\mathcal{G}$, and the outputs are representation vectors for each node, capturing both the feature and structural information of the neighborhood. Most state-of-the-art GNNs use shallow models (*i.e.*, 2 to 3 layers). As first proposed by Li et al. (2018) and further elaborated by Luan et al. (2019); Oono & Suzuki (2020); Zhao & Akoglu (2020); Huang et al. (2020), one of the major challenges to deepen GNNs is the "oversmoothing" of node features — each layer aggregation pushes the neighbor features towards similar values. Repeated aggregation over many layers results in node features being averaged over the full graph. A deep GNN may thus generate indistinguishable embeddings for different nodes. Viewing oversmoothing as a limitation of the layer aggregation, researchers develop alternative architectures. AS-GCN (Huang et al., 2018), DeepGCN (Li et al., 2019) and JK-net (Xu et al., 2018b) use skip-connection across layers. MixHop (Abu-El-Haija et al., 2019), Snowball (Luan et al., 2019) and DAGNN (Liu et al., 2020) enable multi-hop message

passing within a single layer. GraphSAGE (Hamilton et al., 2017a) and GCNII (Ming Chen et al., 2020) encourage self-to-self message passing which effectively form an implicit skip-connection. GIN (Xu et al., 2018a) and DeeperGCN (Li et al., 2020a) propose more expressive neighbor aggregation operations. All the above focus on *architectural exploration*, which is a research direction orthogonal to ours. We can construct the SHADOW version of these GNNs in a plug-and-play fashion. Lastly, DropEdge (Rong et al., 2020) and Bayesian-GDC (Hasanzadeh et al., 2020) propose regularization techniques by adapting dropout (Srivastava et al., 2014) to graphs. Such techniques are only applied during training, and so oversmoothing during inference may not be alleviated.

**Learning from structural information.** Another line of research is to go beyond the layer-wise message passing and more explicitly utilize the graph structural information (Wu et al., 2019; Klicpera et al., 2019a; Bojchevski et al., 2020; Liu et al., 2020; Frasca et al., 2020; You et al., 2019; Li et al., 2020b). In particular, APPNP (Klicpera et al., 2019a) and PPRGo (Bojchevski et al., 2020) utilize the personalized PageRank (Page et al., 1999) algorithm to re-define neighbor connections — instead of propagating features along the (noisy) graph edges, any nodes of structural significance can directly propagate to the target node. Other related methods such as GDC (Klicpera et al., 2019b) and AM-GCN (Wang et al., 2020) reconstructs the adjacency matrix in each GNN layer to short-cut important multi-hop neighbors. Note that all the above methods takes a global view on $\mathcal{G}$ and operate the neural networks on the *full graph*. On the other hand, the idea of using subgraph samples to improve the GNN efficiency has also been explored. For example, SEAL (Zhang & Chen, 2018) extracts local $k$-hop enclosing subgraphs to perform link prediction. GraphSAINT (Zeng et al., 2020) propose random walk samplers to construct minibatches during training.

**Notations.** We focus on the node classification task, although our design principle can be naturally extended to other tasks. Let $\mathcal{G}(\mathcal{V}, \mathcal{E}, \boldsymbol{X})$ be an undirected graph, with node set $\mathcal{V}$, edge set $\mathcal{E} \subseteq \mathcal{V} \times \mathcal{V}$ and node feature matrix $\boldsymbol{X} \in \mathbb{R}^{|\mathcal{V}| \times d}$. The $u$-th row of $\boldsymbol{X}$ corresponds to the length-$d$ feature of node $u$. Let $\boldsymbol{A}$ be the adjacency matrix of $\mathcal{G}$ where $A_{u,v} = 1$ if edge $(u, v) \in \mathcal{E}$ and $A_{u,v} = 0$ otherwise. Denote $\widetilde{\boldsymbol{A}}$ as the adjacency matrix after symmetric normalization (used by GCN), and $\widehat{\boldsymbol{A}}$ as the one after random walk normalization (used by GraphSAGE). Let subscript "$[u]$" mark the quantities corresponding to a small subgraph surrounding node $u$. For example, the subgraph itself is $\mathcal{G}_{[u]}$. For an $L$-layer GNN, let superscript "$(\ell)$" denote the layer-$\ell$ quantities ($1 \leq \ell \leq L$). Let $d^{(\ell)}$ be the number of channels for layer $\ell$; $\boldsymbol{H}^{(\ell-1)} \in \mathbb{R}^{|\mathcal{V}| \times d^{(\ell-1)}}$ and $\boldsymbol{H}^{(\ell)} \in \mathbb{R}^{|\mathcal{V}| \times d^{(\ell)}}$ be the input and output feature matrices. Thus, $\boldsymbol{H}^{(0)} = \boldsymbol{X}$ and $d^{(0)} = d$. Further, let $\boldsymbol{Y} = \boldsymbol{H}^{(L)}$. The operation of a layer can be abstracted as $\boldsymbol{H}^{(\ell)} = f\left(\boldsymbol{H}^{(\ell-1)}, \boldsymbol{A}; \boldsymbol{W}^{(\ell)}\right)$, where $\boldsymbol{W}^{(\ell)}$ are the learnable weights.

## 3 DEEP GNN, SHALLOW SAMPLER

---

**Algorithm 1** SHADOW-GNN inference algorithm

---

**Input:** $\mathcal{G}(\mathcal{V}, \mathcal{E}, \boldsymbol{X})$; Target nodes $\mathcal{V}_t$; GNN model;
**Output:** Node embedding matrix $\boldsymbol{Y}$ for $\mathcal{V}_t$;
1: **for** $v \in \mathcal{V}_t$ **do**
2:     Get $\mathcal{G}_{[v]}\left(\mathcal{V}_{[v]}, \mathcal{E}_{[v]}, \boldsymbol{X}_{[v]}\right)$ by SAMPLE on $\mathcal{G}$
3:     Build $L$-layer GNN with layer operation $f$
4:     $\boldsymbol{y}_v \leftarrow \left[f^L\left(\boldsymbol{X}_{[v]}, \boldsymbol{A}_{[v]}\right)\right]_{v,:}$

---

"Deep GNN, shallow sampler" is a design principle to improve the expressivity and efficiency of GNNs without modifying the layer architecture. Based on this principle, we construct SHADOW-GNN by subgraph sampling. SHADOW-GNN uses the same sampling procedure during both training and inference. The inference algorithm is shown in Algorithm 1. In line 4, we use $f^L$ to denote the composition of the function $f$ for the $L$ layers. Operation $[\cdot]_{v,:}$ slices the $v$-th row of the matrix. We discuss in Section 3.2 how to design SAMPLE to return informative shallow neighborhood.

Notice that the normal GNN is a special kind of SHADOW-GNN. Under the normal setup, the GNN operates on the full graph $\mathcal{G}$ and the $L$ layers propagate the influence from *all* the neighbors up to $L$ hops away from the target node. Such a GNN is equivalent to a SHADOW-GNN when SAMPLE returns the full $L$-hop subgraph. However, following our principle, a good SAMPLE should encourage most of $\mathcal{V}_{[v]}$ to concentrate within the $L_0$-hop neighborhood, where $L_0 < L$. Figures 6, 7 in Appendix show the difference in the neighborhood composition for normal GNN and SHADOW-GNN. Section 3.1 discusses the benefits of making the subgraph shallow and the GNN deep.

### 3.1 ANALYSIS ON EXPRESSIVITY

The SHADOW-GNN design is motivated by the following:

1). A shallow neighborhood is *sufficient* for the GNN to learn a good node representation;

2). A shallow neighborhood is *necessary* to reduce the effect of noise on the GNN;

3). A deep GNN is *necessary* to be expressive on the shallow neighborhood.

Point 1 is true in many real-world scenarios, and is also justified by the $\gamma$-decaying theorem (Zhang & Chen, 2018): to estimate various important graph metrics of a node from its $L$-hop neighborhood, the estimation error decays exponentially with the number of hops $L$. The understanding on Point 2 is two-folded: real-world graphs are likely to include noisy nodes and edges due to erroneous graph construction (Klicpera et al., 2019b). Additionally, even without errors, a node $u$ may still be regarded as "white noise" to the target node $v$, simply because $u$ is far away from $v$ and $u$'s information is irrelevant. The first kind of noise may be handled by filtering out the high frequency components of the node signals (Wu et al., 2019) or enforcing low rank constraints on the adjacency matrix (Grover et al., 2019; Jin et al., 2020). The second kind of noise *cannot be filtered* as long as the GNN aggregates the full $L$-hop neighborhood. In this regard, sampling provides an additional mechanism to deal with both kinds of noises. Also, since we customize different subgraphs for different target nodes, SAMPLE filters noise at the node level rather than the graph level. For Point 3, while theoretical understanding on the benefit of deep neural networks has been well established for Multi-Layer Perceptrons (MLPs) (Telgarsky, 2016), such conclusion does not directly transfer to GNNs. For SHADOW-GNN, we prove a deeper model is more expressive than a shallower one.

**Expressivity comparison with "deep GNN, deep sampler".** When increasing the depth of a normal GNN, the neighborhood around a target node (i.e., the full $L$-hop subgraph) also expands. Eventually, a deep normal GNN on a connected $\mathcal{G}$ will propagate influence of all $\mathcal{V}$ into a single target node. As discussed in Point 2 above, most $\mathcal{V}$ only act as "white noise" to a target node, thus causing difficulties in learning regardless of the layer propagation function.

We pick the GCN architecture for case study. When the GCN depth grows, we analyze how a shallow sampler can filter out the "white noise" in the neighborhood and help preserve target node specific information. Following the setup of Li et al. (2018); Luan et al. (2019); Ming Chen et al. (2020); Zhao & Akoglu (2020), we study the asymptotic behavior of repeatedly applying the neighbor aggregation operation. *i.e.*, we analyze how much information is preserved in $\boldsymbol{M} = \lim_{L \to \infty} \widetilde{\boldsymbol{A}}^L \boldsymbol{X}$.

**Proposition 3.1.** *An $\infty$-layer SHADOW-GCN generates the neighbor aggregation for $v$ as:*

$$\boldsymbol{m}_{[v]} = \left[e_{[v]}\right]_v \cdot \left(\boldsymbol{e}_{[v]}^{\mathsf{T}} \boldsymbol{X}_{[v]}\right) \tag{1}$$

*where $\boldsymbol{e}_{[v]}$ is defined by $\left[e_{[v]}\right]_u = \sqrt{\frac{\delta_{[v]}(u)}{\sum_{w \in \mathcal{V}_{[v]}} \delta_{[v]}(w)}}$; $\delta_{[v]}(u)$ returns the degree of $u$ in $\mathcal{G}_{[v]}$ plus 1.*

OVERSMOOTHING OF NORMAL GCN   With large enough $L$, the full $L$-hop neighborhood becomes $\mathcal{V}$ (assuming connected $\mathcal{G}$). So $\forall \, u, v$, we have $\mathcal{G}_{[u]} = \mathcal{G}_{[v]} = \mathcal{G}$, implying $\boldsymbol{e}_{[u]} = \boldsymbol{e}_{[v]}$ and $\boldsymbol{X}_{[u]} = \boldsymbol{X}_{[v]} = \boldsymbol{X}$. From Proposition 3.1, the aggregation converges to a point where no feature and little structural information of the target is preserved. The only information in $\boldsymbol{m}_{[v]}$ is $v$'s degree.

LOCAL-SMOOTHING OF SHADOW-GCN   With a fixed shallow subgraph, no matter how many times we aggregate using $\widetilde{\boldsymbol{A}}_{[v]}$, the many layers will not include the faraway irrelevant nodes. We can see $\boldsymbol{m}_{[v]}$ as a linear combination of the neighbor node features $\boldsymbol{X}_{[v]}$. Increasing the number of layers only pushes the coefficients of each neighbor features to the stationary values. The domain $\boldsymbol{X}_{[v]}$ of such linear transformation is solely determined by SAMPLE and is independent of the layer depth. Intuitively, if SAMPLE picks non-identical subgraphs for two nodes $u$ and $v$, the aggregations should be different due to the different domains of the linear transformation. Therefore, SHADOW-GCN preserves local feature information whereas normal GCN preserves *none*. For structural information in $\boldsymbol{m}_{[v]}$, note that $\boldsymbol{e}_{[v]}$ is a normalized degree distribution of the subgraph around $v$, and $\left[e_{[v]}\right]_v$ indicates the role of the target node in the subgraph. If we let SAMPLE return the 1-hop subgraph, then $\left[e_{[v]}\right]_v$ alone already contains all the information preserved by a normal GCN, which is $v$'s degree in $\mathcal{G}$. Since $\boldsymbol{e}_{[v]}$ additionally reflects the structure of $v$'s ego-net, a deep SHADOW-GCN with a naïve 1-hop SAMPLE preserves more structural information than a deep GCN.

**Theorem 3.2.** *Let $\overline{m}_{[v]} = \phi_{\mathcal{G}}(v) \cdot m_{[v]}$ where $\phi_{\mathcal{G}}$ is any non-zero function only depending on the structural property of $v$. Let $\mathcal{M} = \{\overline{m}_{[v]} \mid v \in \mathcal{V}\}$. Given $\mathcal{G}$, SAMPLE and some continuous probability distribution in $\mathbb{R}^{|\mathcal{V}| \times f}$ to generate $X$, then $\overline{m}_{[v]} \neq \overline{m}_{[u]}$ if $\mathcal{V}_{[u]} \neq \mathcal{V}_{[v]}$, almost surely.*

**Corollary 3.2.1.** *Consider SAMPLE$_1$ where $\forall v \in \mathcal{V}$, $\left|\mathcal{V}_{[v]}\right| \leq n$. Then $|\mathcal{M}| \geq \left\lceil \frac{|\mathcal{V}|}{n} \right\rceil$ almost surely.*

**Corollary 3.2.2.** *Consider SAMPLE$_2$ where $\forall\, u, v \in \mathcal{V}$, $\mathcal{V}_{[v]} \neq \mathcal{V}_{[u]}$. Then $|\mathcal{M}| = |\mathcal{V}|$ almost surely.*

Theorem 3.2 proves SHADOW-GCN does not oversmooth, as 1). a normal GCN pushes the aggregation of same-degree nodes to the same point, while SHADOW-GCN with SAMPLE$_2$ ensures any two nodes (even with the same degree) can have different aggregation; 2). a normal GCN wipes out all the information in the initial node features after many times of aggregation, while SHADOW-GCN always preserves node feature information. In particular, if we set $\phi_{\mathcal{G}}(v) = \left(\delta_{[v]}(v)\right)^{-1/2}$, a normal GCN generates only one unique value of $\overline{m}$, while SHADOW-GNN can still generate $N$ for any $\phi_{\mathcal{G}}$.

**Expressivity comparison with "shallow GNN, shallow sampler".** Most state-of-the-art GNNs belong to the "shallow GNN, shallow sampler" category. However, we argue that a deep GNN is still desirable even though the neighborhood is shallow. In the following, we first give two examples showing when "shallow GNN, shallow sampler" fails to learn something that a SHADOW-GNN can. Then we prove that SHADOW-GNN is strictly more powerful than 1-dimensional Weisfeiler-Lehman test, thus more powerful than all standard "shallow GNN, shallow sampler" designs.

In Figure 1, consider the 2-hop neighborhood (black dots) of some target node $v$ (blue star). In the first example, suppose another target node $v'$ has a slightly different 2-hop structure as reflected by an additional edge (red dashed line). Assume all nodes have identical features. Then any 2-layer GNN cannot distinguish $v$ and $v'$, while a 3-layer GNN can. The red edge that distinguishes $v$ and $v'$ is not having any influence on $v'$ until three times of message passing. In the second example, we consider a simple task of learning the unweighted mean of neighbor features, $\boldsymbol{\tau} = \frac{1}{|\mathcal{V}_{[v]}|} \sum_{u \in \mathcal{V}_{[v]}} \boldsymbol{x}_u$.

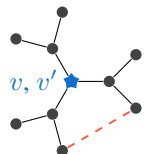

Figure 1: Example neighborhood

Suppose $\mathcal{G}_{[v]}$ consists of the star node, black nodes and black edges. Surprisingly, even with such a "regular" subgraph structure, a normal $L$-layer GraphSAGE cannot learn $\boldsymbol{\tau}$ accurately. The reasons are as follows. If $L > 2$, then normal GraphSAGE will include nodes outside $\mathcal{V}_{[v]}$, and the only way to always exclude the impact of these nodes is by setting the weights for that layer to $0$ — this effectively reduces the number of layers to 2. Now since $\boldsymbol{\tau}$ is linear, ReLU is not desirable, and should be bypassed by shifting $X$ with the bias parameters. The output of the neural net then simplifies to $\boldsymbol{\zeta} = \left[ \sum_{\ell=1}^{L} \widehat{A}_{[v]}^{\ell} X_{[v]} W_\ell \right]_{v,:}$, where $L = 2$ for normal GraphSAGE. For the given $\widehat{A}_{[v]}$, there does not exist a $W_\ell$ for GraphSAGE to satisfy $\boldsymbol{\tau} = \boldsymbol{\zeta}$ for any value of $X_{[v]}$. On the other hand, with SAMPLE returning $\mathcal{G}_{[v]}$, SHADOW-SAGE can learn $\boldsymbol{\tau}$ with any precision if $L$ is allowed to grow. The required $L$ for SHADOW-SAGE to reach the desired precision is determined by the mixing time of $\widehat{A}_{[v]}$. SHADOW-SAGE can learn the unweighted mean because the eigenvector corresponding to the largest eigenvalue of $\widehat{A}_{[v]}$ is a uniform vector. In fact, a deep SHADOW-SAGE can learn the unweighted feature mean on any neighborhood.

**Theorem 3.3.** *Suppose we are given any GNN following the per-layer message passing design as:*

$$h_v^{(\ell)} = f_1^{(\ell)} \left( h_v^{(\ell-1)}, \sum_{u \sim v} f_2^{(\ell)} \left( h_v^{(\ell-1)}, h_u^{(\ell-1)} \right) \right), \qquad (2)$$

*where $f_1^{(\ell)}$ and $f_2^{(\ell)}$ are the layer-$\ell$ update and message functions, respectively, implemented as MLPs; $u \sim v$ denotes $u$ is $v$'s 1-hop neighbor. Then, on the full $L$-hop subgraph, an $L'$-layer GNN is strictly more discriminative than an $L$-layer model, when $L' > L$. Moreover, with injective $f_1$ and $f_2$, the GNN model described above (using $L' > L$ layers on the $L$-hop subgraph) is strictly more discriminative than the 1-dimensional Weisfeiler-Lehman test.*

See Appendix A for the proof. By Theorem 3.3, given a shallow sampler, a deep GNN is strictly more powerful than a shallow one. Even given a deep sampler, we can still use a "deeper" GNN to beat the "deep" GNN. Other practical concerns on deep GNNs include the difficulty in optimization and the computation cost. The optimal depth may be determined after considering such tradeoffs.

## 3.2 SAMPLER DESIGN

$k$**-hop sampler.** Starting from the target node $v$, the sampler traverses up to $k$ hops. At a hop-$\ell$ node $u$, the sampler will add to its hop-$(\ell + 1)$ node set either all neighbors of $u$, or $b$ randomly selected neighbors of $u$. The subgraph $\mathcal{G}_{[v]}$ is induced from all the nodes selected by the sampler. Here depth $k$ and budget $b$ are the sampling parameters. See Appendix C for more analysis.

**PPR sampler.** Personalized PageRank (PPR) has been recently combined with graph learning (Klicpera et al., 2019a; Bojchevski et al., 2020; Li et al., 2020b). However, none of the existing works have explored PPR in the context of subgraph sampling. Given a target $v$, our PPR sampler proceeds as follows: 1). Compute the approximate PPR vector $\boldsymbol{\pi}_v \in \mathbb{R}^{|\mathcal{V}|}$ for $v$. 2). Select the neighborhood $\mathcal{V}_{[v]}$ such that for $u \in \mathcal{V}_{[v]}$, the PPR score $[\pi_v]_u$ is large. 3). Construct the induced subgraph from $\mathcal{V}_{[v]}$. For step 1), even though vector $\boldsymbol{\pi}_v$ is of length-$|\mathcal{V}|$, most of the PPR values are close to zero. So we can use a fast algorithm (Andersen et al., 2006) to compute the approximate PPR score by traversing only the local region around $v$. Throughout the $\boldsymbol{\pi}_v$ computation, most of its entries remain as the initial value of 0. Therefore, the PPR sampler is scalable and efficient w.r.t. both time and space complexity. For step 2), we can either select $\mathcal{V}_{[v]}$ based on top-$p$ values in $\boldsymbol{\pi}_v$, or based on some threshold $[\pi_v]_u > \theta$. Then $p$, $\theta$ are hyperparameters. For step 3), notice that our PPR sampler only uses PPR scores as a node filter. The original graph structure is still preserved among $\mathcal{V}_{[v]}$, due to the induced subgraph step. Such property differentiates our algorithm from existing ones such as APPNP (Klicpera et al., 2019a), GDC (Klicpera et al., 2019b) and PPRGo (Bojchevski et al., 2020). Specifically, the three related works all *reconstruct* the adjacency matrix by directly connecting the target $v$ with nodes of top PPR scores. Reconstruction promotes feature propagation from direct neighbors at the cost of losing structural information, since all $\mathcal{V}_{[v]}$ would become 1-hop away from $v$. We show in Section 3.3 further connections between SHADOW-GNN and the prior arts.

**Extensions.** The $k$-hop and PPR samplers each assume *shortest path distance* and *random walk landing probability* as important metrics for measuring neighbor relevance. Following such rationale, we can design many more sampling algorithms. For example, instead of PPR scores, we can use Katz index (Katz, 1953) as our node filtering condition. This is another way to signify the importance of shortest path distance. If the number of common neighbors is important, we can filter $\mathcal{V}_{[v]}$ by SimRank (Jeh & Widom, 2002). Apart from structural information, `SAMPLE` can also utilize node features. For example, in the $k$-hop sampler, instead of randomly pick $b$ neighbors, we can pick those with highest feature similarity (e.g., by cosine similarity or heat kernel (Wang et al., 2020)).

## 3.3 ARCHITECTURAL EXTENSION

**Subgraph ensemble.** From Section 3.2, each reasonable `SAMPLE` reveals some perspectives of the graph, and they altogether present the full picture. We may then combine the information from various `SAMPLE`. Assume a set of $C$ candidates $\{\text{SAMPLE}_i\}$, each returning $\left(\mathcal{G}_{[v]}\right)_i$. We can either: 1). Construct a union-subgraph $\mathcal{G}_{[v]}^{\star}$ for SHADOW-GNN to operate on, where $\mathcal{V}_{[v]}^{\star} = \bigcup_{i=1}^{C} \left(\mathcal{V}_{[v]}\right)_i$ and $\mathcal{E}_{[v]}^{\star} = \bigcup_{i=1}^{C} \left(\mathcal{E}_{[v]}\right)_i$, or 2). Use $C$ parallel branches of $L$-layer GNN, each operating on $\left(\mathcal{G}_v\right)_i$. So $v$'s output embedding of each branch $i$ (i.e., a vector $\boldsymbol{z}_i$) can be aggregated by attention mechanism:

$$\left(\boldsymbol{Z}_{[v]}\right)_i = f_i^L \left(\left(\boldsymbol{X}_{[v]}\right)_i, \left(\boldsymbol{A}_{[v]}\right)_i\right); \qquad w_i = \text{MLP}\left(\boldsymbol{z}_i\right) \cdot \boldsymbol{q}; \qquad \boldsymbol{y}_v = \sum_{i=1}^{C} \widetilde{w}_i \cdot \boldsymbol{z}_i \qquad (3)$$

where $f_i^L$ is the function by the $L$-layer GNN at branch $i$; $\boldsymbol{z}_i$ is a row vector of $\left(\boldsymbol{Z}_{[v]}\right)_i$ corresponding to $v$; $\boldsymbol{q}$ is a learnable vector; $\widetilde{\boldsymbol{w}}$ is normalized from $\boldsymbol{w}$ by softmax. $\boldsymbol{y}_v$ is the final embedding. We can use identical GNN structure for all the $C$ branches or even share weights among the branches.

$\{\text{SAMPLE}_i\}$ can also perform the same algorithm with different hyperparameters. For example, we can let $\text{SAMPLE}_i$ be the PPR sampler with different threshold $\theta_i$. A SHADOW-SAGE-ensemble can imitate similar feature aggregation behavior as PPRGo (Bojchevski et al., 2020), while still having the additional advantage of learning rich subgraph structural information. Note that PPRGo generates embedding by 1-hop aggregation on the reconstructed graph: $\boldsymbol{\tau}_v = \sum_{u \in \mathcal{V}_{[v]}} \pi_u \boldsymbol{h}_v$, where $\pi_u = [\pi_v]_u$ and $\boldsymbol{h}_v = \text{MLP}\left(\boldsymbol{x}_v\right)$. We can partition $\mathcal{V}_{[v]} = \bigcup_{i=1}^{C} \mathcal{V}_{[v]}^i$ such that nodes in $\mathcal{V}_{[v]}^i$ have similar PPR scores approximated by $\widetilde{\pi}_i$, and $\widetilde{\pi}_i \leq \widetilde{\pi}_{i+1}$. So $\boldsymbol{\tau}_v \approx \sum_{i=1}^{C} \left(\rho_i \sum_{u \in \mathcal{V}_i'} \boldsymbol{h}_u\right)$, where

$\rho_i = \widetilde{\pi}_i - \sum_{j<i} \widetilde{\pi}_j$ and $\mathcal{V}'_i = \bigcup_{k=i}^{C} \mathcal{V}^k_{[v]}$. Recall the discussion on learning unweighted feature mean of any $\mathcal{V}_{[v]}$ (Section 3.1). Setting $\theta_i = \widetilde{\pi}_i$ and $w_i = \rho_i$, SHADOW-SAGE can approximate $\boldsymbol{\tau}_v$. As our PPR sampler also preserves local structure, our model can be more expressive than PPRGo.

**Sampling the reconstructed graph.** Another extension is to build SHADOW-GNN on the graph reconstructed by graph diffusion (Klicpera et al., 2019b; Frasca et al., 2020). In the simplest case, we can apply our $L$-hop sampler on the reconstructed graph, and build a $L'$-layer GNN on the subgraph. When $L = L'$, we recover GDC (Klicpera et al., 2019b). When $L < L'$, we have SHADOW-GDC.

## 4 EXPERIMENTS

**Setup.** We perform node classification on five graphs: `Flickr` (Zeng et al., 2020), `Reddit` (Hamilton et al., 2017a), `Yelp` (Zeng et al., 2020), `ogbn-arxiv` (Hu et al., 2020) and `ogbn-products` (Hu et al., 2020). The graph sizes range from around 9K nodes (`Flickr`) to 2.5M nodes (`ogbn-products`). Consistent with the original setup, we use the metric of "accuracy" for `Flickr`, `Reddit`, `ogbn-arxiv` and `ogbn-products`, and "F1-micro score" for `Yelp`. See Appendix D.1.

We consider seven baselines and construct the corresponding SHADOW models: GCN (Kipf & Welling, 2016), GraphSAGE (Hamilton et al., 2017a), GAT (Veličković et al., 2018), JK-Net (Xu et al., 2018b), GIN (Xu et al., 2018a), SGC (Wu et al., 2019) and GraphSAINT (Zeng et al., 2020). The first six are representatives of the state-of-the-art GNN architectures. They jointly cover various message aggregation functions as well as the skip connection design for facilitating deep GNN training. GraphSAINT is the state-of-the-art minibatch training algorithm for large graphs. The GraphSAINT training has been shown to work well with the GraphSAGE and GAT architectures. In Table 1, for the the rows of GCN, GraphSAGE and GAT, we train 3- and 5-layer models in the full batch fashion. However, for the large graphs, the GPU memory (15GB) is too small to store the full batch. We therefore further run experiments using the GraphSAINT minibatch training algorithm, where "GraphSAINT-RW" denotes the random walk sampler recommended by Zeng et al. (2020).

For all the experiments, we fix the number of channels (*i.e.*, the hidden dimension) for each layer as $d^{(\ell)} = 256$. All accuracy results are measured by five runs without fixing random seeds. Detailed hyperparameter tuning procedure and architecture configuration are described in Appendix D.3.

**Comparison with state-of-the-art.** Table 1 shows the performance comparison of SHADOW-GNN with state-of-the-art GNNs. We use the following metrics: test set accuracy / F1-micro score and inference cost. The inference cost – defined as the average amount of computation to generate prediction for one test node – is purely a measure of computation complexity and is independent of hardware / implementation factors such as parallelization strategy, batch processing, distributed storage, *etc*. Thus, the metric reflects the feasibility of deploying each GNN on real-world workloads, where the graph sizes and hardware specifications can be different from our setup. For SHADOW-GNN, the inference cost does not consider the sampling overhead. See Figure 2 for the sampling time measurement and Appendix B for the equations for calculating the inference cost. For the 3-layer and 5-layer models in Table 1, we simply stack the corresponding GNN layers. During training, we further apply the DropEdge technique (Rong et al., 2020) to both the baseline and SHADOW models. We observe that DropEdge helps improve the baseline accuracy by alleviating oversmoothing, and helps improves the SHADOW accuracy due to its regularization effects.

ACCURACY Comparing the normal GCN, GraphSAGE and GAT, SHADOW-GNNs achieve significantly higher accuracy across all the five datasets. Since we use the identical backbone GNN architectures, the accuracy gains validate the effectiveness of our design principle of "deep GNN, shallow sampler". Note that that our subgraphs are truly shallow, since the the subgraphs generated by our 2-hop and PPR samplers contain no more than 200 nodes. For 3-layer, comparing the accuracy of the normal GNN with the SHADOW-GNN, it is clear that a shallow neighborhood contains sufficient information to generate good node embeddings. For 5-layer accuracy comparison, we observe even higher gains – for normal GNNs, increasing from 3 layers to 5 layers sometimes results in accuracy degradation (even with DropEdge). For SHADOW-GNN, making the GNN deeper is beneficial most of the time. These observations validates our oversmoothing and expressivity analysis in Section 3.1. Finally, the higher accuracy achieved by the PPR sampler compared with the 2-hop sampler demonstrates the importance of a good sampling algorithm.

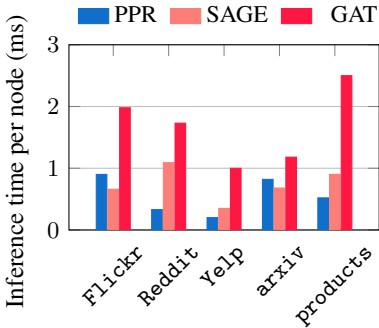

Figure 2: Inference time after parallelization on commodity hardware

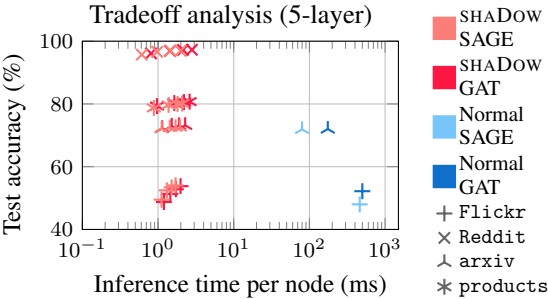

Figure 3: Inference performance tradeoff. We test pretrained models by subgraphs of various sizes.

INFERENCE COST  From Table 1, the inference cost of SHADOW-GNN is often orders of magnitude lower than that of the baselines. The high cost of the baselines reflects the "neighbor explosion" challenge for deep GNNs. In short, for normal GNN, since we do not enforce a fixed size subgraph, the inference computation complexity may grow exponentially with the depth of the GNN (Chen et al., 2017). For example, the number of $\ell$-hop neighbors may be $10\times$ the number of $(\ell-1)$-hop neighbors. For SHADOW-GNN, it is clear that the inference cost only grows linearly with the GNN depth. From Table 1, we observe that even when the GNN depth is only 5, the high computation cost already makes the inference of normal GNNs practically infeasible. The results show the high efficiency of our design. Finally, note that for GraphSAINT, its sampling is only applied during training. The inference computation of GraphSAINT falls back to be identical as its backbone architecture (which samples the full $L$-hop neighborhood). On the other hand, for SHADOW-GNN, we apply the same sampling and batching strategy, whether it is for training or for inference.

Table 1: Comparison on test accuracy / F1-micro score and inference cost tuned with DropEdge)

| Method | Layers | Flickr | | Reddit | | Yelp | | ogbn-arxiv | | ogbn-products | |
| --- | --- | --- | --- | --- | --- | --- | --- | --- | --- | --- | --- |
| | | Accuracy | Cost | Accuracy | Cost | F1-micro | Cost | Accuracy | Cost | Accuracy | Cost |
| GCN | 3 | 0.516±0.002 | 2E0 | 0.953±0.000 | 6E1 | 0.402±0.002 | 2E1 | 0.717±0.003 | 1E0 | 0.756±0.002 | 5E0 |
| | 5 | 0.522±0.002 | 2E2 | 0.949±0.001 | 1E3 | OOM | 1E3 | 0.719±0.002 | 2E0 | OOM | 9E2 |
| GraphSAGE | 3 | 0.514±0.001 | 5E0 | 0.965±0.000 | 9E1 | 0.617±0.003 | 3E1 | 0.719±0.003 | 1E0 | 0.785±0.001 | 8E0 |
| | 5 | 0.515±0.005 | 3E2 | 0.962±0.000 | 2E3 | OOM | 3E3 | 0.719±0.004 | 3E0 | OOM | 2E3 |
| GAT | 3 | 0.507±0.003 | 3E1 | OOM | 5E2 | OOM | 3E2 | 0.720±0.001 | 1E0 | OOM | 6E1 |
| | 5 | 0.516±0.003 | 4E2 | OOM | 3E3 | OOM | 4E3 | OOM | 5E0 | OOM | 4E3 |
| GraphSAGE + GraphSAINT-RW | 3 | 0.517±0.003 | 5E0 | 0.967±0.000 | 9E1 | 0.645±0.001 | 3E1 | 0.710±0.000 | 1E0 | 0.792±0.002 | 8E0 |
| | 5 | 0.520±0.003 | 3E2 | 0.967±0.001 | 2E3 | 0.639±0.000 | 3E3 | 0.701±0.002 | 3E0 | 0.796±0.002 | 2E3 |
| GAT + GraphSAINT-RW | 3 | 0.522±0.005 | 3E1 | 0.967±0.000 | 5E2 | 0.645±0.000 | 3E2 | 0.697±0.000 | 1E0 | 0.802±0.003 | 6E1 |
| | 5 | 0.515±0.003 | 4E2 | 0.965±0.002 | 3E3 | 0.647±0.001 | 4E3 | 0.695±0.001 | 5E0 | 0.799±0.007 | 4E3 |
| SHADOW-GCN +PPR | 3 | 0.526±0.002 | (1) | **0.958**±0.000 | (1) | 0.526±0.000 | (1) | 0.719±0.002 | (1) | 0.777±0.003 | (1) |
| | 5 | **0.527**±0.002 | 1E0 | **0.958**±0.000 | 1E0 | **0.527**±0.001 | 2E0 | **0.721**±0.001 | 2E0 | **0.784**±0.003 | 2E0 |
| SHADOW-SAGE + 2-hop | 3 | 0.529±0.001 | 2E0 | 0.966±0.000 | 2E0 | 0.649±0.000 | 2E0 | 0.716±0.001 | 2E0 | 0.799±0.001 | 2E0 |
| | 5 | 0.534±0.004 | 3E0 | 0.966±0.000 | 3E0 | 0.650±0.000 | 3E0 | 0.718±0.001 | 3E0 | 0.801±0.002 | 3E0 |
| SHADOW-SAGE + PPR | 3 | 0.534±0.002 | 2E0 | **0.969**±0.000 | 2E0 | **0.651**±0.000 | 2E0 | 0.723±0.003 | 3E0 | 0.794±0.002 | 2E0 |
| | 5 | **0.542**±0.002 | 3E0 | **0.969**±0.000 | 3E0 | 0.650±0.000 | 3E0 | **0.725**±0.001 | 3E0 | **0.804**±0.002 | 3E0 |
| SHADOW-GAT + PPR | 3 | **0.538**±0.003 | 2E0 | 0.970±0.001 | 2E0 | **0.655**±0.000 | 2E0 | 0.724±0.001 | 3E0 | 0.801±0.001 | 2E0 |
| | 5 | 0.534±0.002 | 3E0 | **0.971**±0.001 | 3E0 | 0.654±0.000 | 3E0 | **0.728**±0.003 | 5E0 | **0.809**±0.003 | 3E0 |

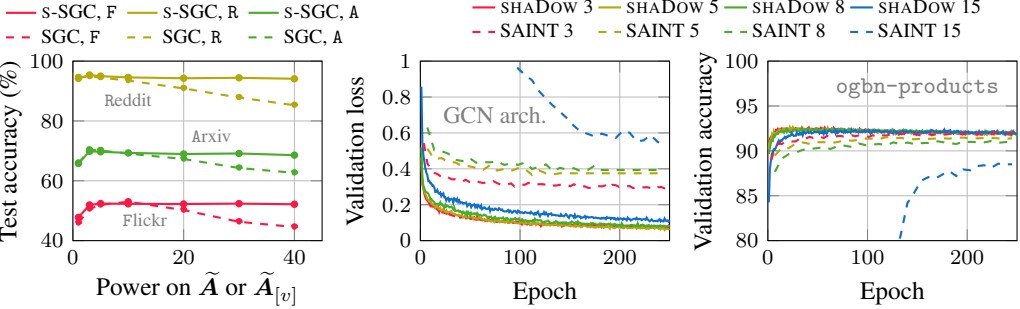

Figure 4: Effect of increasing the model depth on the SGC and GCN architectures

**Evaluation on PPR sampler.** We evaluate the PPR sampler in terms of its execution time overhead and accuracy-time tradeoff. In Figure 2, we parallelize the sampler on a 40-core Xeon CPU and the GNN computation on a NVIDIA P100 GPU (See Appendix D.2). The sampler is lightweight: the sampling time is lower than the GNN computation time in most cases. Also, the sampling time per node does not necessarily grow with the full graph size. By Section 3.2, the approximate PPR computation achieves efficiency and scalability by only traversing a local region around each target node. To evaluate the accuracy-time tradeoff, we take the 5-layer models of Table 1 as the pretrained models. Then for SHADOW-GNN, we vary the PPR budget $p$ from 50 to 200 with stride 50. In Figure 3, the inference time of SHADOW-GNN has already included the PPR sampling time. Firstly, consistent with Table 1, inference of SHADOW-GNNs achieves higher accuracy than the normal GNNs, with orders of magnitude speedup as well. In addition, based on the application requirements (*e.g.*, latency constraint), SHADOW-GNNs have the flexibility of adjusting the sampling size without the need of retraining. For example, on `Reddit` and `ogbn-arxiv`, directly reducing the subgraph size from 200 to 50 speeds up inference by $2\times$ to $4\times$ at the cost of less than $1\%$ accuracy drop.

**Evaluation on other architectures.** In addition to the GCN, GraphSAGE and GAT models in Table 1, we further compare JK-Net and GIN with their SHADOW versions in Table 2. Similar to DropEdge, the skip connection (or "jumping knowledge") helps

Table 2: Test accuracy (%) on other architectures

|  | Flickr | | Reddit | | ogbn-arxiv | |
|---|---|---|---|---|---|---|
|  | Normal | SHADOW | Normal | SHADOW | Normal | SHADOW |
| JK (3) | 49.45±0.70 | **53.17**±0.27 | 96.49±0.10 | **96.82**±0.03 | 71.30±0.26 | **72.01**±0.17 |
| JK (5) | 49.40±0.83 | **53.28**±0.26 | 96.40±0.13 | **96.85**±0.06 | 71.66±0.53 | **72.26**±0.24 |
| GIN (3) | 51.32±0.31 | **52.28**±0.28 | 93.45±0.34 | **95.78**±0.06 | 70.87±0.16 | **71.73**±0.29 |
| GIN (5) | 50.04±0.67 | **52.55**±0.23 | 75.50±0.39 | **95.52**±0.07 | 69.37±0.62 | **71.40**±0.27 |

accuracy improvement on deeper models. Compared with the normal JK-Net, increasing the depth benefits SHADOW-JK more. The GIN architecture theoretically does not oversmooth. However, we observe that the GIN training is very sensitive to hyperparameter settings. We hypothesize that such a challenge is due to the sensitivity of the sum aggregator on noisy neighbors (*e.g.*, for GraphSAGE, a single noisy node can hardly cause a significant perturbation on the aggregation, due to the averaging over the entire neighborhood). The accuracy improvement of SHADOW-GIN compared with the normal GIN may thus be due to noise filtering by shallow sampling (see Section 3.1). The impact of noises / irrelevant neighbors can be critical, as reflected by the 5-layer GIN accuracy on `Reddit`.

**Effect of model depth.** To validate Theorem 3.2 on the non-oversmoothing of SHADOW-GCN, we experiment on much larger GNN depth. In the left plot of Figure 4, we run a normal SGC and a SHADOW-SGC by varying the power on the adjacency matrix from 1 to 40 (see Appendix D.4 for the model details and setup). While SGC gradually collapses local information into global "white noise", accuracy of SHADOW-SGC

Table 3: GCN test accuracy (%)

| $L'$ | SAMPLE | Flickr | ogbn-products |
|---|---|---|---|
| 3 | PPR | 52.57±0.21 | 77.73±0.32 |
| 5 | 2-hop | 52.10±0.23 | 77.94±0.39 |
|  | PPR | 52.73±0.20 | 78.36±0.34 |
|  | Ensemble | **53.04**±0.17 | **78.58**±0.21 |
| 7 | PPR | 52.25±0.23 | **78.50**±0.44 |

does not degrade. In the middle and right plots, we train the standard GCN architecture without the SGC simplifications. Since `ogbn-products` is too large, we compare SHADOW-GCN with GCN-SAINT. For both methods, the deeper models converge slower. Deeper models (whether they are GNNs or other types of NNs) are generally harder to optimize. For SHADOW-GCN, the validation performance of the 15-layer model eventually catches up with the shallower ones. For GCN-SAINT, however, there consistently exists a large performance gap between the 15-layer model and the shallower ones. Both experiments clearly show that shallow sampling prevents oversmoothing. Finally, for `ogbn-products` in in Table 3, we note that even through $> 98\%$ of the subgraph nodes are with 2 hops (see Figure 6), increasing the GNN depth can benefit accuracy for up to $L' = 7$ layers.

**Evaluation on subgraph ensemble.** Benefit of ensemble is shown via the 5-layer entries in Table 3. Different samplers, such as PPR and $k$-hop, preserve different kinds of local information. Rather than designing one "perfect" sampler, it may be more reasonable to ensemble a few simpler ones. Note that ensemble is only possible for subgraph based SHADOW-GNN. In Figure 8 (see Appendix E), we show that even the simplest 1-hop sampler can benefit training after ensemble.

## 5 CONCLUSION

We have presented a design principle, "deep GNNs, shallow samplers", enabling accurate and efficient inference for a wide range of GNN architectures. We have theoretically justified the expressivity of SHADOW-GNN and empirically demonstrated its benefits under various setups and metrics.

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

# A PROOFS

*Proof of Proposition 3.1.* The GCN model performs symmetric normalization on the adjacency matrix. SHADOW-GCN follows the same way to normalize the subgraph adjacency matrix as:

$$\widetilde{\boldsymbol{A}}_{[v]} = \left(\boldsymbol{D}_{[v]} + \boldsymbol{I}\right)^{-\frac{1}{2}} \cdot \left(\boldsymbol{A}_{[v]} + \boldsymbol{I}\right) \cdot \left(\boldsymbol{D}_{[v]} + \boldsymbol{I}\right)^{-\frac{1}{2}} \tag{4}$$

where $\boldsymbol{A}_{[v]} \in \mathbb{R}^{N \times N}$ is the binary adjacency matrix for $\mathcal{G}_{[v]}$.

$\widetilde{\boldsymbol{A}}_{[v]}$ is a real symmetric matrix and has the largest eigenvalue of 1. Since SAMPLE ensures the subgraph $\mathcal{G}_{[v]}$ is connected, so the multiplicity of the largest eigenvalue is 1. By Theorem 1 of Huang et al. (2020), we can bound the eigenvalues $\lambda_i$ by $1 = \lambda_1 > \lambda_2 > \ldots > \lambda_N > -1$.

Performing eigen-decomposition on $\widetilde{\boldsymbol{A}}_{[v]}$, we have

$$\widetilde{\boldsymbol{A}}_{[v]} = \boldsymbol{E}_{[v]} \boldsymbol{\Lambda} \boldsymbol{E}_{[v]}^{-1} = \boldsymbol{E}_{[v]} \boldsymbol{\Lambda} \boldsymbol{E}_{[v]}^{\mathsf{T}} \tag{5}$$

where $\boldsymbol{\Lambda}$ is a diagonal matrix $\Lambda_{i,i} = \lambda_i$ and matrix $\boldsymbol{E}_{[v]}$ consists of the $N$ normalized eigenvectors. We have:

$$\widetilde{\boldsymbol{A}}_{[v]}^{L} = \boldsymbol{E}_{[v]} \boldsymbol{\Lambda}^{L} \boldsymbol{E}_{[v]}^{\mathsf{T}} \tag{6}$$

Since $|\lambda_i| < 1$ when $i \neq 1$, we have $\lim_{L \to \infty} \widetilde{\boldsymbol{A}}_{[v]}^{L} = \boldsymbol{e}_{[v]} \boldsymbol{e}_{[v]}^{\mathsf{T}}$, where $\boldsymbol{e}_v$ is the eigenvector corresponding to $\lambda_1$. It is easy to see that $\left[\boldsymbol{e}_{[v]}\right]_u \propto \sqrt{\delta_{[v]}(u)}$ (Huang et al., 2020). After normalization, $\left[\boldsymbol{e}_{[v]}\right]_u = \sqrt{\frac{\delta_{[v]}(u)}{\sum_{w \in \mathcal{V}_{[v]}} \delta_{[v]}(w)}}$.

It directly follows that $\boldsymbol{m}_{[v]} = \left[\boldsymbol{e}_{[v]}\right]_v \cdot \left(\boldsymbol{e}_{[v]}^{\mathsf{T}} \boldsymbol{X}_{[v]}\right)$, with value of $\boldsymbol{e}_{[v]}$ defined above. $\square$

*Proof of Theorem 3.2.* We first prove the case of $\overline{\boldsymbol{m}}_{[v]} = \boldsymbol{m}_{[v]}$. *i.e.*, $\phi_{\mathcal{G}}(v) = 1$.

According to Proposition 3.1, the aggregation for each target node equals $\boldsymbol{m}_{[v]} = \left[\boldsymbol{e}_{[v]}\right]_v \boldsymbol{e}_{[v]}^{\mathsf{T}} \boldsymbol{X}_{[v]}$. Let $N = |\mathcal{V}|$. Let $\bar{\bar{\boldsymbol{e}}}_{[v]} \in \mathbb{R}^{N \times 1}$ be the vector expanded from $\boldsymbol{e}_{[v]}$, such that $\left[\bar{\bar{\boldsymbol{e}}}_{[v]}\right]_u = 0$ if $u \notin \mathcal{V}_{[v]}$ and the rest of the non-zero elements of $\bar{\bar{\boldsymbol{e}}}_{[v]}$ are copied from $\boldsymbol{e}_{[v]}$ accordingly.

We define:

$$\boldsymbol{\epsilon} = \left[\boldsymbol{e}_{[v]}\right]_v \cdot \bar{\bar{\boldsymbol{e}}}_{[v]}^{\mathsf{T}} - \left[\boldsymbol{e}_{[u]}\right]_u \cdot \bar{\bar{\boldsymbol{e}}}_{[u]}^{\mathsf{T}} \tag{7}$$

Suppose we only compute $\boldsymbol{\epsilon}$ when two nodes $u$ and $v$ have non-identical sampled neighborhood. *i.e.*, $\mathcal{V}_{[u]} \neq \mathcal{V}_{[v]}$. Then given $\mathcal{G}$, SAMPLE and some distribution to generate $\boldsymbol{X}$, there are finite number of possible $\boldsymbol{\epsilon}$ values. The reason is that $\mathcal{G}$ is finite and $\boldsymbol{\epsilon}$ does not depend on $\boldsymbol{X}$ (where $\boldsymbol{X}$ itself can take infinitely many values). Each of the possible $\boldsymbol{\epsilon}$ values defines a hyperplane in $\mathbb{R}^{N-1}$ by $\boldsymbol{\epsilon} \cdot \boldsymbol{x} = \boldsymbol{0}$ (where $\boldsymbol{x} \in \mathbb{R}^N$). Let all such hyperplanes be $\mathcal{H}$.

Now suppose we build SHADOW-GCN by performing SAMPLE on $\mathcal{G}$ and $\boldsymbol{X}$. For any two $v$ and $u$ with $\mathcal{V}_{[v]} \neq \mathcal{V}_{[u]}$, suppose their aggregations are the same: $\boldsymbol{m}_{[v]} = \boldsymbol{m}_{[u]}$. Then we must have $\boldsymbol{\epsilon} \cdot \boldsymbol{X} = \boldsymbol{0}$. In other words, $\forall i$, $\boldsymbol{X}_{:,i}$ must fall on one of the hyperplanes in $\mathcal{H}$.

However, since $\boldsymbol{X}$ is generated from a continuous distribution in $\mathbb{R}^{N \times f}$, $\boldsymbol{X}_{:,i}$ would not fall on all of the hyperplanes in $\mathcal{H}$, almost surely. Therefore, for any $v$ and $u$ such that $\mathcal{V}_{[v]} \neq \mathcal{V}_{[u]}$, $\boldsymbol{m}_{[v]} \neq \boldsymbol{m}_{[u]}$ almost surely.

For a more general $\phi_{\mathcal{G}}(v)$, since $\phi_{\mathcal{G}}$ does not depend on $\boldsymbol{X}$, the proof follows exactly the same steps as above.

$\square$

*Proof of Corollary 3.2.1.* Note that $\forall u \in \mathcal{V}$, $u \in \mathcal{V}_{[u]}$. For any node $v$, there are at most $n-1$ other nodes in $\mathcal{V}$ with the same neighborhood as $\mathcal{V}_{[v]}$. Such $n-1$ possible nodes are exactly those in $\mathcal{V}_{[v]}$.

By Theorem 3.2, $\forall v \in \mathcal{V}$, there are at most $n - 1$ other nodes in $\mathcal{V}$ having the same aggregation as $\boldsymbol{m}_{[v]}$. Equivalently, total number of possible aggregations is at least $\lceil |\mathcal{V}|/n \rceil$. □

*Proof of Corollary 3.2.2.* By definition of SAMPLE₂, any pair of nodes have non-identical neighborhood. By Theorem 3.2, any pair of nodes have non-identical aggregation. Equivalently, all nodes have different aggregation and $|\mathcal{M}| = |\mathcal{V}|$. □

*Proof of Theorem 3.3.* Define $\mathcal{G}^L_{[v]}$ as the subgraph induced from all the $\ell$-hop neighbors of $v$, where $1 \le \ell \le L$.

We first prove that an $L'$-layer GNN is at least as expressive as an $L$-layer GNN on any $L$-hop subgraph. We note that for the target node $v$, the only difference between these two architectures is that an $L$-layer GNN exactly performs $L$ message passing iterations to propagate node information from at most $L$ hops away, while an $L'$-layer GNN has $L' - L$ more message passing iterations before performing the $L$ message passings. Thanks to the universal approximation theorem (Hornik et al., 1989), we can let $f_1^{(\ell)}\left(\boldsymbol{h}_v^{(\ell-1)}, \sum_{u \sim v} f_2^{(\ell)}\left(\boldsymbol{h}_v^{(\ell-1)}, \boldsymbol{h}_u^{(\ell-1)}\right)\right) = \boldsymbol{h}_v^{(\ell-1)}, \forall 1 \le \ell \le L' - L$ since MLPs can model and learn such functions. Then, the $L'$-layer GNN will have the same output as the $L$-layer GNN.

Then, we show that an $L'$-layer GNN can learn something that an $L$-layer GNN cannot learn on some $\mathcal{G}^L_{[v]}$. This can be proved exactly by the example given in Figure 1, where the red edge is included in the subgraph. In this $\mathcal{G}^2_{[v']}$ around $v'$, a 2-layer GNN fails to capture the red edge, and will output the representation of $v'$ the same as when there is no red edge. In contrast, a GNN with more than 2 layers can capture the red edge immediately after the first message passing round, where the two end nodes of the red edge receive messages from each other. And after 3 message passing rounds, this captured additional information will be propagated to $v'$, resulting in a representation different from that learned by the 2-layer GNN.

Finally, we prove that with injective $f_1$ and $f_2$, the GNN model described above (using $L' > L$ layers on the $L$-hop subgraph) is strictly more discriminative than the 1-dimensional Weisfeiler-Lehman (1-WL) test. We first prove that no matter how many iterations $L$ 1-WL runs, we can always run this GNN model on the $L$-hop subgraph with $L'$ layers to get at least the same discriminative power as 1-WL. According to Xu et al. (2018a), an $L$-layer GNN on $\mathcal{G}^L_{[v]}$ is theoretically as discriminative as 1-WL by being able to discriminate all nodes $v$ that 1-WL with $L$ iterations of color update can discriminate. Suppose there are two node $v, v'$ that 1-WL gives different colors to after $L$ iterations. Then applying the $L$-layer GNN on $\mathcal{G}^L_{[v]}$ and $\mathcal{G}^L_{[v']}$ will also return $\boldsymbol{h}_v^{(L)} \ne \boldsymbol{h}_{v'}^{(L)}$. Since $f_1$ and $f_2$ are injective, we know $\boldsymbol{h}_v^{(L+i)} \ne \boldsymbol{h}_{v'}^{(L+i)}$ for any $i > 0$, which means an $L'$-layer GNN ($L' > L$) can also discriminate the two nodes.

Then it suffices to give an example where 1-WL always fails to distinguish two nodes no matter how many iterations it runs, while the GNN model above can discriminate them with some $L$ and $L'$.

We construct a very simple example, as shown in Figure 5. Consider an undirected graph with two connected components (CC). The first CC is a hexagon. The second CC is a triangle. Both CC are 2-regular. Suppose all nodes have identical attributes and edges are unweighted. Our goal is to discriminate a node $u$ in the first CC from a node $v$ in the second CC. For 1-WL, no matter how many iterations $L$ it runs, 1-WL always assigns the same label to $u$ and $v$. For the GNN above, when we set $L = 1$ and $L' = 2$, we can already discriminate between the two nodes.

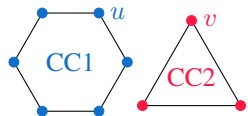

Figure 5: Example graph on which SHADOW-GNN is more expressive than 1-WL

□

**Remark** Note that even though we discuss the $(L + 1)$-layer case in the above proof, increasing the layer $L'$ beyond $L + 1$ can still be benefitial in many cases. Consider the following:

CASE 1 We revisit the GraphSAGE example on Figure 1. Suppose we use a $L'$-layer GraphSAGE to approximate a function $\tau$ on the subgraph $\mathcal{G}^L_{[v]}$. We consider an example $\tau$ as computing the unweighted mean of the subgraph node features. In this case, the error of approximating $\tau$ converges

to zero when $L'$ goes to infinity. The error can still be significant if $L'$ is not sufficiently larger than $L$. The desired depth $L'$ is determined by the mixing time of the subgraph adjacency matrix.

CASE 2   Consider a more "powerful" GNN when $f_1$ and $f_2$ of Equation 2 perform injective mapping. Recall the following two facts:

- On the full graph, such a GNN of $L_0$ layers is as discriminative as the 1-WL test running $L_0$ iterations.
- 1-WL may take up to $O(N)$ iterations to converge where $N$ is the full graph size.

Now extend to the SHADOW-GNN case. Consider two target nodes $u$ and $v$ with their corresponding $L$-hop subgraphs $\mathcal{G}^L_{[u]}$ and $\mathcal{G}^L_{[v]}$. An $L'$-layer GNN will output different embeddings for $u$ and $v$, if $u$'s color assigned by $L'$-iteration 1-WL on $\mathcal{G}^L_{[u]}$ is different from $v$'s color assigned by $L'$-iteration 1-WL on $\mathcal{G}^L_{[v]}$. The coloring of $u$ and $v$ can keep changing for $O(N)$ iterations (here $N$ is the size of $\mathcal{G}^L_{[u]}$ and $\mathcal{G}^L_{[v]}$). Thus, increasing the GNN depth on $L$-hop subgraphs can improve discriminative power for up to $O(N)$ layers.

## B   INFERENCE COMPLEXITY CALCULATION

Here we describe the equations to compute the "inference cost" of Table 1. Recall that inference cost is a measure of computation complexity to generate node embeddings for a given GNN architecture.

The numbers in Table 1 shows on average, how many arithmetic operations is required to generate the embedding for each node. For a GNN layer $\ell$, denote the number of input nodes as $n^{(\ell-1)}$ and the number of output nodes as $n^{(\ell)}$. Denote the number of edges connecting the input and output nodes as $m^{(\ell)}$. Recall that we use $d^{(\ell)}$ to denote the number of channels, or, the hidden dimension.

In the following, we ignore the computation cost of non-linear activation, batch-normalization and applying bias, since their complexity is negligible compared with the other operations.

For the GCN architecture, each layer mainly performs two operations: aggregation of the neighbor features and linear transformation by the layer weights. So the number of multiplication-addition (MAC) operations of a layer equals:

$$C^{(\ell)}_{\text{GCN}} = m^{(\ell)}d^{(\ell-1)} + n^{(\ell)}d^{(\ell-1)}d^{(\ell)} \tag{8}$$

Similarly, for GraphSAGE architecture, the number of MAC operations equals:

$$C^{(\ell)}_{\text{SAGE}} = m^{(\ell)}d^{(\ell-1)} + 2 \cdot n^{(\ell)}d^{(\ell-1)}d^{(\ell)} \tag{9}$$

where the 2 factor is due to the weight transformation on the self-features.

For GAT, suppose the number of attention heads is $t$. Then the layer contains $t$ weight matrices $\boldsymbol{W}^i$, each of shape $d^{(\ell-1)} \times \frac{d^{(\ell)}}{t}$. We first transform each of the $n^{(\ell-1)}$ nodes by $\boldsymbol{W}^i$. Then for each edge $(u, v)$ connecting the layer input $u$ to the layer output $v$, we obtain its edge weight (*i.e.*, a scalar) by computing the dot product between $u$'s, $v$'s transformed feature vectors and the model attention weight vector. After obtaining the edge weight, the remaining computation is to aggregate the $n^{(\ell-1)}$ features into the $n^{(\ell)}$ nodes. The final output is obtained by concatenating the features of different heads. The number of MAC operations equals:

$$C^{(\ell)}_{\text{GAT}} = t \cdot n^{(\ell-1)}d^{(\ell-1)}\frac{d^{(\ell)}}{t} + 2t \cdot m^{(\ell)}\frac{d^{(\ell)}}{t} + m^{(\ell)}d^{(\ell)} \tag{10}$$

$$= 3m^{(\ell)}d^{(\ell)} + n^{(\ell-1)}d^{(\ell-1)}d^{(\ell)} \tag{11}$$

On the same graph, GCN is less expensive than GraphSAGE. GraphSAGE is in general less expensive than GAT, due to $n^{(\ell-1)} > n^{(\ell)}$. In addition, we note that for all architectures, $C_*^{(\ell)}$ grows proportionally with $n^{(\ell)}$ and $m^{(\ell)}$. For the normal GNN architecture, since we are using the full $\ell$-hop neighborhood for each node, the value of $n^{(\ell)}$ and $m^{(\ell)}$ may grow exponentially with $\ell$. This is the "neighbor explosion" phenonemon and is the root cause of the high inference cost of the Table 1 baselines.

For SHADOW-GNN, suppose the subgraph contains $n$ nodes and $m$ edges. Then $n^{(\ell)} = n$ and $m^{(\ell)} = m$. The inference cost of any SHADOW-GNN is ensured to only grow linearly with the depth of the GNN.

## C   DETAILS ON SAMPLERS

Here we present more detailed discussion on the proposed $k$-hop and PPR samplers.

$k$**-hop sampler.**   While previous works Hamilton et al. (2017a); Ying et al. (2018) have proposed to randomly sub-sample the multi-hop neighborhood to improve GNN computation efficiency, none of them explicitly set the GNN depth to be larger than $k$.

For example, GraphSAGE also uses a $k$-hop sampler. In their case, the sampling improves the computation speed at the cost of accuracy drop. In our case, such a sampling process can lead to both accuracy gain and computation efficiency improvement – This is partially due to the exclusion of noisy neighbors by the sampler, and partially due to the compensation of the information loss by the higher expressivity of deeper models. More importantly, in the GraphSAGE design, the sampling depth $k$ grows with the model depth. Thus, "neighbor explosion" is inevitable unless the sampling budget $b$ is set to 1 (in such a case, the sampler will likely return a path, which does not contain much information).

Another difference between our $k$-hop sampler and that of GraphSAGE is that SHADOW-GNN runs on the *induced subgraph* of the $k$-hop neighbors. Compared with the normal GraphSAGE, a SHADOW-SAGE is able to discover more complicated structural information from the subgraph constructed by our $k$-hop sampler (See Theorem 3.3).

**PPR sampler.**   In Sections 3.1 and 3.3, we have made detailed comparison between PPR-based SHADOW-GNN and related works such as Klicpera et al. (2019a;b); Bojchevski et al. (2020). Here we would like to highlight that the approximate PPR algorithm proposed in Andersen et al. (2006) is both scalable and efficient. The number of nodes we need to visit to obtain the approximate PPR vector is much smaller than the graph size. In addition, each visit only involves a scalar update, which is orders of magnitude cheaper than the cost of neighbor propagation in a GNN. Quantitatively, for the three largest datasets, `Reddit`, `Yelp` and `ogbn-products`, for each target node on average, the number of nodes touched by the PPR computation is comparable to the full 2-hop neighborhood size, as shown in Table 4. The statistics of Table 4 also explains the empirical execution time measurement presented in Figure 2.

Table 4: Average number of nodes touched by the approximate PPR computation

| Dataset | Average 2-hop size | Average # nodes touched by PPR |
|---|---|---|
| Reddit | 11093 | 27751 |
| Yelp | 2472 | 5575 |
| ogbn-products | 3961 | 5405 |

## D   DETAILED EXPERIMENTAL SETUP

### D.1   ADDITIONAL DATASET DETAILS

The statistics for the five benchmark graphs is listed in Table 5. Note that for `Yelp`, each node may have multiple labels, and thus we follow the original paper (Zeng et al., 2020) to report its

F1-micro score. For all the other graphs, a node is only associated with a single label, and so we report accuracy. Note that for `Reddit` and `Flickr`, other papers (Hamilton et al., 2017a; Zeng et al., 2020) also report F1-micro score as the metric. However, since each node only has a single label, "F1-micro score" is exactly the same as "accuracy".

Table 5: Dataset statistics

| Dataset | Setting | Nodes | Edges | Degree | Feature | Classes | Train / Val / Test |
|---|---|---|---|---|---|---|---|
| Flickr | Inductive | 89,250 | 899,756 | 10 | 500 | 7 | 0.50 / 0.25 / 0.25 |
| Reddit | Inductive | 232,965 | 11,606,919 | 50 | 602 | 41 | 0.66 / 0.10 / 0.24 |
| Yelp | Inductive | 716,847 | 6,977,410 | 10 | 300 | 100 | 0.75 / 0.10 / 0.15 |
| ogbn-arxiv | Transductive | 169,343 | 1,166,243 | 7 | 128 | 40 | 0.54 / 0.18 / 0.29 |
| ogbn-products | Transductive | 2,449,029 | 61,859,140 | 25 | 100 | 47 | 0.10 / 0.02 / 0.88 |

### D.2 HARDWARE SPECIFICATION AND ENVIRONMENT

We run our experiments on a DGX1 machine with Dual 20-core Intel Xeon CPUs (E5-2698 v4 @ 2.2Ghz) and eight NVIDIA Tesla P100 GPUs (15GB of HBM2 memory). The main memory is 512GB DDR4 memory. The code is written in `Python 3.6.8` (where the sampling part is written with `C++` parallelized by `OpenMP`, and the interface between `C++` and `Python` is via `PyBind11`). We use `PyTorch 1.4.0` on `CUDA 10.2` with `CUDNN 7.2.1` to train the model on GPU.

When measuring the execution time of Figure 2 and 3, we use the CPU (40-cores in total) to sample subgraphs in parallel and the GPU to execute the GNN layer operations.

### D.3 HYPERPARAMETER SEARCHING PROCEDURE

For all the experiments, as described in the "setup" paragraph of Section 4, we set the hidden dimension to be always $d^{(\ell)} = 256$. In addition, for all the GAT and SHADOW-GAT experiments, we set the number of attention heads to be $t = 4$. For all the GIN and SHADOW-GIN experiments, we use a 2-layer MLP (with hidden dimension 256) to perform the injective mapping in each GIN layer. For all the JK-Net and SHADOW-JK experiments, we use the concatenation operation to aggregate the hidden features of each layer in the JK layer.

For all the baseline and SHADOW-GNN experiments, we use Adam optimizer (Kingma & Ba, 2014). We perform grid search on the hyperparameter space defined by:

- Activation function: $\{\texttt{ReLU}, \texttt{ELU}\}$
- Dropout: $\{0.00, 0.05, 0.10, 0.15, 0.20, 0.25, 0.30, 0.35, 0.40, 0.45, 0.50\}$
- DropEdge: $\{0.00, 0.05, 0.10, 0.15, 0.20, 0.25, 0.30, 0.35, 0.40, 0.45, 0.50\}$
- Learning rate: $\{0.01, 0.002, 0.001, 0.0002, 0.0001, 0.00002\}$

The sampling hyperparameters are tuned as follows.

For the PPR sampler, we consider two versions: one based on fixed sampling budget $p$ and the other based on PPR score thresholding $\theta$.

- If with fixed budget, we then we disable the $\theta$ thresholding. We tune the budget by $\theta \in \{100, 125, 150, 175, 200\}$.
- If with thresholding, we set $\theta \in \{0.01, 0.05\}$. We still have an upper bound $p$ on the subgraph size. So if there are $q$ nodes in the neighborhood with PPR score larger than $\theta$, the final subgraph size would be $\max\{p, q\}$. Such an upper bound eliminates the corner cases which may cause hardware inefficiency due to very large $q$. In this case, we set $p$ to be either 200 or 500 depending on the graph.

For the $k$-hop sampler, we define the hyperparameter space as:

- Depth $k = 2$

Table 6: Training configuration of SHADOW-GNN for Table 1 (PPR sampler)

| Arch. | Dataset | Layers | Learning Rate | Dropout | DropEdge | Budget $p$ | Threshold $\theta$ |
|---|---|---|---|---|---|---|---|
| SHADOW-GCN | Flickr | 3 | 0.001 | 0.40 | 0.10 | 200 | – |
| | | 5 | 0.001 | 0.40 | 0.10 | 200 | – |
| | Reddit | 3 | 0.00002 | 0.20 | 0.05 | 150 | – |
| | | 5 | 0.00002 | 0.20 | 0.05 | 150 | – |
| | Yelp | 3 | 0.001 | 0.10 | 0.00 | 100 | – |
| | | 5 | 0.001 | 0.10 | 0.00 | 100 | – |
| | ogbn-arxiv | 3 | 0.00002 | 0.25 | 0.10 | 200 | – |
| | | 5 | 0.00002 | 0.25 | 0.10 | 200 | – |
| | ogbn-products | 3 | 0.002 | 0.40 | 0.05 | 150 | – |
| | | 5 | 0.002 | 0.40 | 0.05 | 150 | – |
| SHADOW-SAGE | Flickr | 3 | 0.001 | 0.40 | 0.00 | 200 | – |
| | | 5 | 0.001 | 0.40 | 0.00 | 200 | – |
| | Reddit | 3 | 0.0002 | 0.20 | 0.10 | 150 | – |
| | | 5 | 0.0002 | 0.20 | 0.10 | 150 | – |
| | Yelp | 3 | 0.001 | 0.10 | 0.00 | 200 | – |
| | | 5 | 0.001 | 0.10 | 0.00 | 200 | – |
| | ogbn-arxiv | 3 | 0.0001 | 0.30 | 0.10 | 500 | 0.01 |
| | | 5 | 0.00002 | 0.25 | 0.15 | 200 | 0.01 |
| | ogbn-products | 3 | 0.002 | 0.40 | 0.15 | 150 | – |
| | | 5 | 0.002 | 0.40 | 0.15 | 150 | – |
| SHADOW-GAT | Flickr | 3 | 0.001 | 0.40 | 0.00 | 200 | – |
| | | 5 | 0.001 | 0.40 | 0.00 | 200 | – |
| | Reddit | 3 | 0.0001 | 0.20 | 0.00 | 150 | – |
| | | 5 | 0.0001 | 0.20 | 0.00 | 150 | – |
| | Yelp | 3 | 0.001 | 0.10 | 0.00 | 100 | – |
| | | 5 | 0.001 | 0.10 | 0.00 | 100 | – |
| | ogbn-arxiv | 3 | 0.0001 | 0.20 | 0.00 | 175 | – |
| | | 5 | 0.0001 | 0.20 | 0.00 | 175 | – |
| | ogbn-products | 3 | 0.001 | 0.35 | 0.00 | 150 | – |
| | | 5 | 0.001 | 0.35 | 0.00 | 150 | – |

Table 7: Training configuration of SHADOW-GNN for Table 1 ($k$-hop sampler)

| Arch. | Dataset | Layers | Learning Rate | Dropout | DropEdge | Budget $b$ | Depth $k$ |
|---|---|---|---|---|---|---|---|
| SHADOW-SAGE | Flickr | 3 | 0.001 | 0.40 | 0.00 | 20 | 2 |
| | | 5 | 0.001 | 0.40 | 0.00 | 20 | 2 |
| | Reddit | 3 | 0.0001 | 0.20 | 0.00 | 15 | 2 |
| | | 5 | 0.0001 | 0.20 | 0.00 | 15 | 2 |
| | Yelp | 3 | 0.01 | 0.10 | 0.00 | 5 | 2 |
| | | 5 | 0.01 | 0.10 | 0.00 | 5 | 2 |
| | ogbn-arxiv | 3 | 0.0001 | 0.20 | 0.00 | 10 | 2 |
| | | 5 | 0.0001 | 0.20 | 0.00 | 10 | 2 |
| | ogbn-products | 3 | 0.002 | 0.35 | 0.00 | 10 | 2 |
| | | 5 | 0.002 | 0.35 | 0.00 | 10 | 2 |

- Budget $b \in \{5, 10, 15, 20\}$

The hyperparameters to reproduce the Table 1 SHADOW-GNN results are listed in Tables 6 and 7. The code will be shared with the reviewers at the end of the rebuttal period and released to public after paper acceptance.

### D.4 SETUP OF THE EXPERIMENTS ON DEEPER MODELS

**SGC and SHADOW-SGC.** Following Wu et al. (2019), we compute the SGC model as $Y = \texttt{softmax}\left(\widetilde{A}^K X W\right)$ where $\widetilde{A} = \widetilde{D}^{-\frac{1}{2}} \widetilde{A} \widetilde{D}^{-\frac{1}{2}}$ and $\widetilde{A} = I + A$. Matrix $W$ is the only learnable parameter. $K$ is the power on the adjacency matrix and we vary it as $K \in \{1, 3, 5, 10, 20, 30, 40\}$ in the Figure 4 experiments. For SHADOW-SGC, according to Algorithm 1, we compute the embedding for target $v$ as $y_v = \left[\texttt{softmax}\left(\widetilde{A}_{[v]}^K X_{[v]} W\right)\right]_{v,:}$.

SGC and SHADOW-SGC are trained with the same hyperparameters (*i.e.*, learning rate equals 0.001 and dropout equals 0.1, across all datasets). SHADOW-SGC uses the same sampler as the SHADOW-GCN model in Table 1. In the legend of Figure 4, due to lack of space, we use S-SGC to denote SHADOW-SGC. We use "F", "R" and "A" to denote the datasets of `Flickr`, `Reddit` and `ogbn-arxiv` respectively.

**SHADOW-GCN and GCN-SAINT.** For SHADOW-GCN and GCN-SAINT each, we first tune the 5-layer version to obtain the best-performing hyperparameter settings. Then we fix such hyperparameters and change the number of layers to up to 15. Note that GraphSAINT also performs subgraph sampling (during training only). As specified in the "Setup" of Section 4, we use the random-walk sampler of GraphSAINT. We further tune the GraphSAINT sampling parameters to optimize its convergence quality. We obverse that GraphSAINT requires a relatively large batch size to perform well (*i.e.*, number of subgraph nodes is around 20,000). This is also consistent with the configuration for `ogbn-products` in the official GraphSAINT repository.

One more thing to notice is that in the official GraphSAINT implementation, it evaluates the validation set performance as follows:

1). Propagate all the GNN layers using the full graph adjacency matrix (including all the training, validation and test nodes);

2). Compute loss on the full graph. Use this loss as an approximation of the validation loss.

3). Mask out the predictions on the validation nodes and compute the validation accuracy.

Such a way of loss computation is due to the specific implementation design choices of Graph-SAINT. In the GraphSAINT official repository, it has clarified that such a way of computing the validation loss is not very accurate. However, since the validation loss is neither used for back-propagation nor as the stopping criteria, such an approximate loss computation does not affect the optimization procedure at all. On the other hand, the impact to our experiments is that for the middle plot of Figure 4, the scale of the GraphSAINT loss curves may need to be adjusted to better reflect the actual validation loss values. Note that such an adjustment, if performed, would equally affect all the GraphSAINT curves for different number of layers. Therefore, we can still conclude that increasing the number of layers have a significant negative effect on the GCN-SAINT convergence (this conclusion can also be confirmed by the right plot of Figure 4, where the validation accuracy of GCN-SAINT is *not* obtained by approximation).

## E ADDITIONAL EXPERIMENTAL RESULTS

### E.1 IS THE PPR SAMPLER REALLY "SHALLOW"?

The PPR sampler defines the neighborhood $\mathcal{V}_{[v]}$ based on the magnitude of the PPR scores $\widetilde{\pi}_v$. Although it is possible that a node far away from the target $v$ may have a high PPR score and be selected by SAMPLE, such a case rarely happens in practice. Most of the nodes in $\mathcal{V}_{[v]}$ are concentrated within the 2-hop neighborhood of $v$. Let $\texttt{dist}\,(v, u)$ denote the shortest path distance between nodes $u$ and $v$. Figure 6 shows the distribution of $\texttt{dist}\,(v, u)$ for $u \in \mathcal{V}_{[v]}$.

On the other hand, Figure 7 shows the composition of the neighborhood under the normal GNN. It is clear that most of the nodes propagated by the normal GNN are far away from the target node.

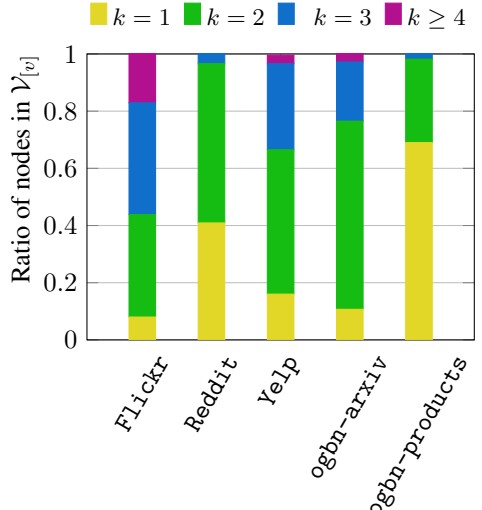 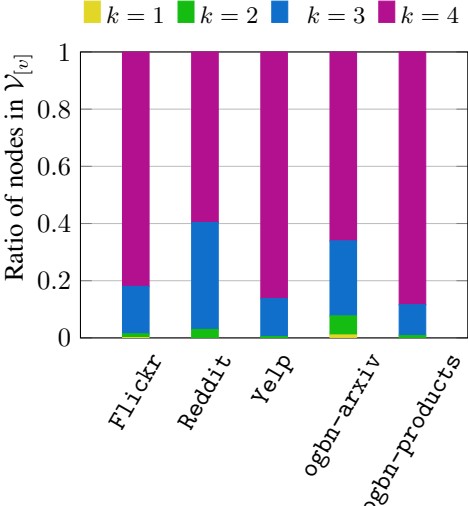

Figure 6: PPR sampler for 5-layer SHADOW-GNN: Composition of nodes $u$ in $\mathcal{V}_{[v]}$ under various $\text{dist}\,(u,v) = k$

Figure 7: 4-hop sampler for normal 4-layer GNN: Composition of nodes $u$ in $\mathcal{V}_{[v]}$ under various $\text{dist}\,(u,v) = k$

### E.2  EXAMPLE ON SUBGRAPH ENSEMBLE

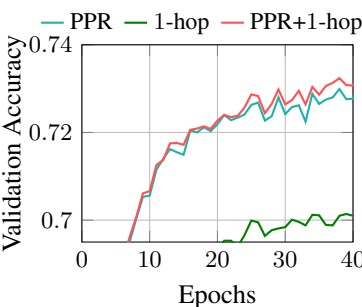

Figure 8: Subgraph ensemble: SHADOW-SAGE on `ogbn-arxiv`

In addition to the Table 3 results on SHADOW-GCN, Figure 8 further shows how subgraph ensemble improves convergence quality with the SHADOW-SAGE model. We consider a PPR sampler with fixed threshold $\theta = 0.01$, a 1-hop sampler (which returns the full 1-hop neighbors) and the ensemble of these two. The 1-hop sampler incurs significant information loss since it preserves no multi-hop neighbors. As expected, its accuracy is low. Surprisingly, we observe accuracy gain when we ensemble the 1-hop sampler and the PPR sampler by Equation 3. As discussed, different samplers preserve different kinds of local information. Rather than trying to design one "perfect" sampler, it may be more reasonable to ensemble a few simpler ones.

