# OpenReview forum: "Deep Graph Neural Networks with Shallow Subgraph Samplers"
_ICLR.cc/2021/Conference — Reject_

### Official Review · AnonReviewer1 · 2020-10-28
**Review of Deep Graph Neural Networks with Shallow Subgraph Samplers**

**Rating:** 5
**Confidence:** 2

**Review:**

This paper proposed a simple technique to improve the performance of message passing neural network models. The proposed idea is relatively simple, instead of applying GNN to the entire graphs, which leads to the over smoothing problem with enough number of layers, for each node, the model samples an ego-network of the target node, from which GNN can model representation of the node. The experimental results show the competitive results of the proposed method while preventing the over smoothing issue. Below, I listed some of my concerns about the paper:

- Theorem 3.3 seems incorrect. The statement ' L'-layer GNN is strictly more discriminative than the 1-WL test' seems incorrect. If you run WL test with L iteration, this statement seems okay, however, in general, when you run the WL test, the minimum number of iteration is not set to L. The paper referred to in the proof also mentioned that only GNN is theoretically as discriminative as 1-WL when the depth of the layers is sufficient enough [Xu et al. 2018a]

- The paper emphasizes the necessity of deep architecture whereas in experiments 3-layer and 5-layer architectures, which seem not very deep, are used. It would be interesting to see how the performance changes as one increase the depth of layers.

- Experimental setup: Ablation study shows the best performance when multiple samplers are used, whereas the main results do not contain the result of the ensemble.

- Consistency between table1 and table 2: In table 1 2-hop sampler is used to show the best performance while in the ablation study 1-hop sampler is used. Is there a reason why different k-hop samplers are used in these cases?

---

> ### Author Response · Authors · 2020-11-17
> **Initial Reply to Reviewer 1: Clarification on Conceptual Concerns**
>
> Thanks for the thoughtful reviews. We appreciate the feedback.
>
> While the general principle of "deep GNN, shallow sampler" is simple, the various theoretical analysis to justify it, together with the sampling algorithms and architecture extensions to realize it, are all highly non-trivial. We believe the intuition behind shaDow-GNN is simple yet "non-conventional".
>
> Before addressing your specific concerns, we would like to briefly reiterate the significance of our method. You are right that oversmoothing is one of the most important motivations for shaDow-GNNs, and constructing ego-nets is one of the example samplers mentioned in the paper. Beyond that, our analysis and designs throughout the whole paper highlights the *generality* and *flexibility* of shaDow in a broader sense:
> * shaDow-GNN improves expressive power for a wide class of GNN architectures, ranging from spectral based (e.g., GCN [Theorem 3.2, as acknowledged by the reviewer]), spatial based (e.g., GraphSAGE [example of learning the $\tau$ function], Section 3.1), generally injective mapping based (e.g., GIN [revised Theorem 3.3, see also discussion below]) to graph reconstruction based (e.g., PPRGo, GDC [two extensions of Section 3.3]).
> * shaDow-GNN can integrate various graph sampling algorithms in a plug-and-play fashion. In addition to the mentioned ego-net, k-hop and PPR samplers, we also discuss the principles of designing new samplers and list a few other candidate algorithms in the "Extension" paragraph of Section 3.2. We leave the detailed evaluation of other samplers as future work.
>
> In the following, we address your concerns.
>
> -----------------------------------------------------------------
> ### Regarding the correctness of Theorem 3.3
>
> We apologize for the confusion raised by the original theorem statement. We argue that Theorem 3.3 is still correct: i.e., shaDow-GNN is more expressive than 1-WL, even when 1-WL is allowed to run infinite iterations.
>
> The main idea of the proof is mostly similar to the one presented in the Appendix of our original paper. However, we construct a different example graph to better illustrate discriminative power of shaDow-GNN:
>
> The first step of the proof is to show that shaDow-GNN is at least as powerful as 1-WL. We consider an $L'$-layer shaDow-GNN on the $L$-hop subgraph, with $L' > L$ and injective $f_1$ and $f_2$. Suppose we run 1-WL for $T$ iterations, where $T$ can range from $1$ to $\infty$. We can always set $L=T$ and $L' > L$ to make such shaDow-GNN as discriminative as the 1-WL. Such an argument is similar to the one shown in the original proof.
>
> The second step is to show there exists a case where shaDow-GNN can discriminate something that 1-WL cannot discriminate. We construct a very simple example. Consider an undirected graph with two connected components (CC). The first CC is a hexagon. The second CC is a triangle. Both CC are 2-regular. Suppose all nodes have identical attributes and edges are unweighted. Our goal is to discriminate a node $u$ in the first CC from a node $v$ in the second CC.
> * For 1-WL, no matter how many iterations it runs, 1-WL always assigns the same color / label to $u$ and $v$.
> * For shaDow-GNN, when we set $L=1$ and $L'=2$, we can already discriminate between the two nodes.
>
> In summary, the above proves the correctness of Theorem 3.3 -- shaDow is more discriminative than 1-WL, even when 1-WL is allowed to run many iterations. Note that for this Theorem, we allow the parameters $L$, $L'$ and $T$ to vary freely to reach the maximum discriminative power of shaDow-GNN and 1-WL, respectively.
>
> We will revise the Theorem 3.3 in our next revision.
>
> --------------------------------------------------------------------------------------
> ### Regarding the experiments on GNNs deeper than 5 layers
>
> We are conducting such additional experiments, and we will follow up with this thread to update the results.
>
> ----------------------------------------------
> ### Regarding the ablation study
>
> The part of ablation study (Figure 2 in the original submission) is self-contained. It is not performing ablation on the results in Table 1. In other words, we demonstrate the effectiveness of ensemble by comparing the convergence of "PPR alone", "1-hop alone", with "PPR + 1-hop". Our message is that, even though the sampler under ensemble is very lossy (i.e., 1-hop is the simplest possible sampler), the ensemble architecture can still lead to high quality learning. The intuitive explanation behind this observation is that each sampled subgraph preserves different characteristics of the target node.
>
> We will add additional results on ensemble in our revision.
>
> --------------------------------------
> Thanks again for your review.

---

> > ### Author Response · Authors · 2020-11-23
> > **Summary of Revision 1**
> >
> > We have uploaded our first revision of the paper, focusing on additional experiments regarding sampler and architecture evaluation. In the next revision, we will include the results on GNNs deeper than 5 layers and more results on subgraph ensemble. We have observed empirical evidence that shaDow-GCN do not oversmooth.
> >
> > ### Additional experimental results
> >
> > **[3 more shaDow architectures]**
> >
> > We have implemented and evaluated shaDow-GCN (Table 1), shaDow-GIN (Table 2) and shaDow-JK (Table 2). All the additional architectures demonstrate the benefit of our shaDow construction by significantly higher accuracy than the baselines.
> >
> > **[Detailed evaluation on sampling cost]**
> >
> > We have included the detailed evaluation of sampling overhead in the added Figure 2.
> >
> > **[Ablation on sampling parameters]**
> >
> > In the added Figure 3, we evaluate the effect of PPR sampling size on the accuracy and inference time. We have observed tradeoff between the two metrics. In summary, the shaDow construction allows us to flexibility adjust the sampling size without the need of retraining.
> >
> > **[Deeper models]**
> >
> > We are currently wrapping up this set of experiments. We will include the results in the next revision.
> >
> > ### Additional technical results
> >
> > **[Theorem 3.3]**
> >
> > We have updated the statement and proof of Theorem 3.3, reflecting our previous post.
> >
> > ----------------------------
> >
> > Please let us know of any concerns. Thanks a lot!

---

> > > ### Author Response · Authors · 2020-11-24
> > > **Summary of Revision 2: Additional Experimental Results**
> > >
> > > We have included thorough evaluation on deeper GNN models as well as subgraph ensemble.
> > >
> > > ### Deeper models
> > >
> > > We show the effect of deeper models by three perspectives:
> > >
> > > * **Convergence speed**: We have presented the convergence curves for GCN and shaDow-GCN under 3, 5, 8 and 15 layers. Results show that shallow sampling significantly improves the convergence quality (both the rate and final accuracy). The deeper the model is, the more benefits from the shallow sampler.
> > > * **Test accuracy**: One of the original review is that 3 and 5 layers do not seem very deep. In our opinion, the model depth is a concept relative to the subgraph property. For example, for the ogbn-products graph, 98% of the subgraph nodes fall within the 2-hop neighborhood, and therefore 7, 5, or even 3 layers can be considered as deep. In the added Table 3, we find that increasing the model to up to 7 layers can still lead to accuracy improvement, comparing to the 3-layer and 5-layer models.
> > > * **SGC architecture**: The SGC architecture performs the same aggregation as GCN. Yet it omits activations. So we think it as a good model to demonstrate oversmoothing and validate Theorem 3.2. We have implemented and evaluated shaDow-SGC. We increase the power on the adjacency matrix from 1 to 40, and results clearly show that shaDow-SGC achieves significantly higher accuracy, especially with higher power (corresponding to deeper layers).
> > >
> > >
> > > ### Evaluation on subgraph ensemble
> > >
> > > We have included more results to demonstrate the benefit of ensemble. In the added Table 3, ensemble of PPR and 1-hop improves test accuracy compared with PPR alone or 1-hop alone. Taking into account the reviewer's suggestion, the new ensemble results are regarding 2-hop and PPR (rather than 1-hop) samplers.
> > >
> > > ----------------------------------------------------
> > > As the rebuttal period is ending soon, we thank you again for your valuable feedback. We believe we have addressed all of your concerns in the revision.

---

### Official Review · AnonReviewer2 · 2020-10-28
**Good theoretical insights but a weak empirical validation of proposed theories**

**Rating:** 5
**Confidence:** 4

**Review:**


To address the oversmoothing problem and reduce the computational cost of GNNs, this paper proposes to train deep GNNs with shallow subgraph samplers. The following two theoretical proofs provide insightful motivations of Shadow-GNN: (1 )Obtaining node embeddings within shallow subgraphs can avoid oversmoothing; (2)Deep GNNs are strictly more expressive than a shallow one. Experiments are performed on five different graph datasets with 3 and 5 layer GNNs coupling with a k-hop sampler or a Personalized PageRank (PPR) sampler.

Pros:

+ The paper is well-presented including nice illustrations and good examples;
+ Theorem 3.2 and Theorem 3.3 provide nice motivations of designing shallow subgraph samplers and deep GNNs respectively;
+ Variants of subgraph samplers and architectural extensions are discussed thoroughly, e.g., k-hop sampler, PPR sampler, subgraph ensemble and sampling the reconstructed graph.

Cons:

- Regarding Theorem 3.3 and the example in  Figure 1, It seems for a L-hop subgraph L+1-layer GNN has enough expressivity. In other words, do we need a deeper L’-layer GNN with  L’ > L+1? Or does a L’-layer GNN with  L’ > L+1 strictly more expressive a L+1-layer GNN? If not, it can not prove we need models deeper than L+1 layers for learning on a L-hop subgraph.
- Proposition 3.1 and Theorem 3.2 show a normal GNN suffers from over-smoothing as the number of layers approaches infinity but SHADOW-GNN does not. However, they are not well validated in the experiments for the following two reasons: (1) the assumed GNN formulate $\mathbf{M} = \text{lim}_{L\to \infty} \widetilde{\mathbf{A}}^{L} \mathbf{X}$ is very different from SAGE  and GAT used in the experiments; (2) the number of layers experimented are 3 and 5 layers which are far from "deep" compared to the infinite layers condition. Thus, it would be great to have experiments to bridge the gap between the theory and the practice. For example,  training a SGC (Wu et al., 2019) model with more than 20 layers to show the effectiveness of  Shadow-GNN would be interesting.
- Only comparing the inference cost of one node is not fair. For Shadow-GNN, each node will need to build one subgraph for the inference. This seems to be very computationally heavy. It would be great if there is a comparison of the inference time on the whole graph.

Minor Comment:
* The terms of Shadow-GNN and Shadow-GCN are used inconsistently.

References:
* Wu, F., Souza Jr, A.H., Zhang, T., Fifty, C., Yu, T. and Weinberger, K.Q., 2019, January. Simplifying Graph Convolutional Networks. In ICML.


================================

Post rebuttal Comments:

Thank the authors for the response. After reading the rebuttal, I decided to change my rating to 5 for the following reasons:

* Theorem 3.3 in the original manuscript has a major flaw which is also mentioned in the review of reviewer 3. The remedied theorem is considered as a substantial change compared to the original one.
* Experiments do not fully justify the superiority of the proposed method in training deep GNNs. The performance on ogbn-product degrades when the depth of GNNs goes to 15. It is not clear whether shallow subgraph samplers really help with training "deep" models.
* The empirical results are not strong enough to support the theoretical claims. The best results achieved by shaDow-GNN on ogbn-arxiv and ogbn-product are 72.28 and 80.09 which rank 15th and 9th respectively on the OGB leaderboard. The results should be improved before being accepted.

---

> ### Author Response · Authors · 2020-11-17
> **Initial Reply to Reviewer 2: Clarification on Conceptual Concerns (1/2)**
>
> Thank you for the thorough review. We appreciate your positive feedback on our theoretical insights into shaDow-GNNs. Here we would like to first clarify the conceptual understanding of the paper. Afterwards, we would follow up with additional empirical results. We would also reflect the discussion in this thread in the next revision.
>
> ----------
> ### Re. concern on Theorem 3.3
>
> Theoretically, there are still benefits of shaDow-GNN beyond $L+1$ layers. In the original version, we only discuss the $L'=L+1$ case in Appendix since it suffices to prove the Theorem 3.3 statement. Here we explain why making $L'$ much larger than $L$ is still theoretically desirable.
>
> (Note, the following part between the && markers is overlapping with the reply to Reviewer 3)
>
> &&
>
> **[Case 1]**: Suppose we use a $L'$-layer GraphSAGE to approximate a function $\tau$ on the subgraph $G_{[u]}^L$. We consider an example $\tau$ as computing the unweighted mean of the subgraph node features. In this case, the error of approximating $\tau$ converges to zero when $L'$ goes to infinity. The error can still be significant if  $L'$ is not sufficiently larger than $L$ . The $L'$ for a desired error rate is determined by the mixing time of the subgraph adjacency matrix. See the discussion on the second example regarding Figure 1.
>
> **[Case 2]**: Consider a more "powerful" GNN when $f_1$ and $f_2$ of Equation 3 perform injective mapping. Recall the following:
> * On the full graph, such a GNN of $L_0$ layers is as discriminative as the 1-WL test running $L_0$ iterations.
> * 1-WL may take up to $O(N)$ iterations to converge where $N$ is the full graph size.
>
> Now extend to the shaDow case. Consider two target nodes $u$ and $v$ with their corresponding $L$-hop subgraphs $G_{[u]}^L$ and $G_{[v]}^L$. An $L'$-layer GNN will output different embeddings for $u$ and $v$, if $u$'s color assigned by $L'$-iteration 1-WL on $G_{[u]}^L$ is different from $v$'s color assigned by $L'$-iteration 1-WL on $G_{[v]}^L$. The coloring of $u$ and $v$ can keep changing for $O(N)$ iterations (here $N$ is the size of $G_{[u]}^L$ and $G_{[v]}^L$). Thus, increasing the GNN depth on $L$-hop subgraphs can improve discriminative power for up to $O(N)$ layers. Empirically, the subgraph size $N$ is set between 100 to 200.
>
> So from the above theoretical perspective, on a shallow subgraph, we may see the benefit of deepening GNN for a truly large depth. In Theorem 3.2, since the statement only specifies a general condition of $L' > L$, it suffices to discuss just the $L'=L+1$ case in the proof. On the other hand, we agree with you that discussing the benefit beyond $L'=L+1$ would clarify the understanding. We will add the above analysis to the paper.
>
> &&
>
> -------------
> ### Re. empirical validation of Proposition 3.1 and Theorem 3.2
>
> You are right that experiments on shaDow-SAGE and shaDow-GAT may not be regarded as an empirical validation of non-oversmoothing of shaDow-GCN. We are conducting additional experiments with shaDow-GCN and will include its results in the next revision. Your recommendation of using shaDow-SGC is also highly valuable. We will also include such results.
>
> On the other hand, we would like to clarify that the theoretical analysis on the shaDow design is not limited to the GCN type of spectral convolution. In addition to Proposition 3.1 and Theorem 3.2, we have also analyzed the expressivity gain over
> * GraphSAGE, via the example of learning the $\tau$ function (Figure 1, Section 3.1).
> * generally injective mapping based GNNs (e.g., GIN), via (revised) Theorem 3.3.
> * graph reconstruction based GNNs (e.g., PPRGo, GDC), via the two architectural extensions shown in Section 3.3.
>
> The current experimental results on shaDow-SAGE thus is still closely connected to the above theoretical analysis. We will include results on shaDow-GIN and (potentially) shaDow-GDC in the next revision.
>
> In summary, we will show additional experiments to fully backup the various theoretical claims in the paper.
>
> --------------------
>
> (Please continue checking part 2 of this response)

---

> > ### Author Response · Authors · 2020-11-17
> > **Initial Reply to Reviewer 2: Clarification on Conceptual Concerns (2/2)**
> >
> > ---------
> > ### Re. concern on the sampling cost
> >
> > While it is true that each node needs to build up its own subgraph, such a step does *not* invalidate our claim on inference efficiency:
> > * Both the proposed samplers (k-hop, PPR) are localized and scalable. For PPR, thanks to the approximate algorithm proposed by Anderson et al., 2006 (see Section 3.2), the number of nodes we need to visit to obtain the approximate PPR vector is much smaller than the graph size. In addition, each visit only involves a scalar update, which is orders of magnitude cheaper than the cost of neighbor propagation in a GNN. Quantitatively, for the three largest datasets, Reddit, Yelp and ogbn-products, for each target node on average, the number of nodes touched by the PPR computation is comparable to the full 2-hop neighborhood size:
> >
> > | Dataset | Average 2-hop size |  Average # nodes touched by PPR  |
> > |:---------|:---------------------:|:---------------------:|
> > | Reddit | 11093 | 27751 |
> > | Yelp | 2472 | 5575 |
> > | ogbn-products | 3961 | 5405 |
> >
> >
> > * Secondly, each node can construct its own subgraph completely independently. This makes it trivial to perform parallelization and batch processing. In addition, shaDow-GNNs can use exactly the same parallelization strategy for the training and inference phases. Such a property does not hold for the other methods -- Recall that state-of-the-art minibatch strategies such as GraphSAINT does not apply to the inference phase.
> > * In many practical applications, the graph structure is evolving at a much slower pace than the node attributes (e.g, in a social network, the users may keep posting new contents without affecting friendship). In such a scenario, while we still need to frequently generate/regenerate the embeddings for the target users, we can reuse the fixed subgraphs constructed before.
> >
> > We will perform thorough evaluation on the inference time (including the GNN computation time and the subgraph construction time) on the real machine. We will provide such results in the revision.
> >
> > -----------------------
> > ### Re. minor comments
> >
> > We would like to clarify that we intentionally distinguish the terms of shaDow-GNN and shaDow-GCN. Specifically, "shaDow-GNN" in our paper is a general term. It can refer to different GNN architectures. E.g., shaDow-GCN, shaDow-SAGE, shaDow-GAT all belong to the shaDow-GNN family. The term shaDow-GCN only refers to the shaDow version of the GCN architecture.
> >
> > ------------------------------
> > Thanks again for your helpful comments. We will keep updating this thread during the rebuttal period.

---

> > > ### Author Response · Authors · 2020-11-23
> > > **Summary of Revision 1**
> > >
> > > We have uploaded our first revision of the paper. In the next revision, we will include the results on GNNs deeper than 5 layers and more results on subgraph ensemble. We have observed empirical evidence that shaDow-GCN do not oversmooth.
> > >
> > > ------------------------
> > >
> > > ### Additional experimental results
> > >
> > > **[Detailed inference cost analysis]**
> > >
> > > We have performed thorough evaluation on the sampler cost during inference. The added Figure 2 shows the actual execution time of the PPR sampler as compared to the GNN layer computation time on GPU. Consistent with the algorithmic analysis, the proposed sampler is both efficient and scalable.
> > >
> > > We have also included the total inference time by considering both the sampling / subgraph building time and the layer computation time in the added Figure 3. In addition, Figure 3 also shows the accuracy-time tradeoff by varying the subgraph size. The flexibility of adjusting the sample sizes of shaDow-GNNs without the need of retraining better fits the shaDow models to many real-life applications.
> > >
> > > **[Other shaDow architectures]**
> > >
> > > We have implemented and evaluated three more architectures: shaDow-GCN (Table 1), shaDow-GIN (Table 2) and shaDow-JK (Table 2). All three architectures verify the benefits of the shaDow construction from the accuracy and efficiency perspectives. In particular, the GIN vs. shaDow-GIN comparison empirically shows the benefit of shallow sampling from the perspective of excluding noises.
> > >
> > > **[Evaluation on deeper models]**
> > >
> > > This set of experiments is still running, and we have seen empirical evidence that shaDow models do not oversmooth. We will wrap up the results and present them in the next revision.
> > >
> > > ### Additional Technical Materials
> > >
> > > We have modified Theorem 3.3 and its proof based on the review and the previous post.
> > >
> > > ----------------------------
> > >
> > > Please let us know of any other concerns. Thanks!

---

> > > > ### Author Response · Authors · 2020-11-24
> > > > **Summary of Revision 2: Additional Results on Deeper Models and Ensemble**
> > > >
> > > > One of the important feedback from the reviewer is to validate the non-oversmoothing of shaDow models. The review gave a very good suggestion of comparing SGC and shaDow-SGC. We have included such results in this revision. Together with other evaluation on deep models, we believe we have gathered sufficient evidence that shaDow models do not oversmooth.
> > > >
> > > > ### Results on deep models
> > > >
> > > > We evaluate deep models:
> > > >
> > > > * **shaDow-SGC**: We have implemented shaDow-SGC and compared it with the normal SGC. We vary the power from 1 to up to 40. For normal SGC, we observe significant accuracy degradation beyond 10 layers, while for shaDow-SGC, such a degradation does not happen. This result matches the conclusion of Theorem 3.2 well.
> > > > * **Convergence quality**: We have presented the convergence curves for GCN and shaDow-GCN under 3, 5, 8 and 15 layers. Results show that shallow sampling significantly improves the convergence quality (both the rate and final accuracy). The deeper the model is, the more benefits from the shallow sampler.
> > > > * **Test accuracy**: We have presented the test accuracy for models of 3, 5 and 7 layers. Interestingly, on the ogbn-products graph, even though over 98% of the subgraph nodes fall within 2 hops, increasing the model to up to 7 layers can still lead to accuracy improvement.
> > > >
> > > > ### Results on ensemble
> > > >
> > > > We have presented additional results on subgraph ensemble in the added Table 3.
> > > >
> > > > --------------------------------------
> > > > We think your valuable feedback indeed helps us improve our paper. We believe all of the concerns have been addressed.
> > > >
> > > > Thanks again!

---

### Official Review · AnonReviewer4 · 2020-10-28
**The paper presents a shallow graph sampler combined with a deep graph neural network framework to train large graphs. Overall, this method seems well-motivated, and both theoretical and empirical results support their claims. There are few poins on which clarification from the authors would be helpful.**

**Rating:** 7
**Confidence:** 3

**Review:**

This paper proposes a new extension of GNNs to deep GNNs, which use subgraphs to keep the computational costs low for training large graphs. It addresses the two main reasons that GNNs have not previously been extended to deep GNNs: expressivity and computational cost. Increasing the number of layers in a GNN leads to averaging over more nodes, which in turn collapses the learned embeddings. The paper claims that using shallow graphs instead of the full graphs avoids this oversmoothing issue. Additionally, using the full graph is computationally expensive since the neighborhood sizes grow with the number of neighbors. Using shallow subgraphs instead allows the size of the neighborhoods to remain constant as the number of layers increase. To this end, the paper presents SHADOW-GNN, a Deep GNN with shallow sampling. They extend this framework to GraphSAGE and GAT models and show that it improves performance over the original model with a lower computational cost. Overall, this method seems well-motivated, and both theoretical and empirical results support their claims. There are a few points on which clarification from the authors would be helpful.

Strengths:
++ The paper presents 3 motivations - (1) shallow neighborhood is sufficient to learn graph representation (2) it is necessary to reduce over smoothing issues (3) One still needs a deep GNN model to learn effectively form the shallow neighborhood - and it supports these claims by providing examples and formal proofs in the form of Proposition 3.1, Theorem 3.2, and Theorem 3.3, respectively.

++ The paper recommends using two main samplers for sampling the shallow neighborhoods of a node - (1) $k$-hop sampler, which randomly selects $b$ neighbors and (2) Personalized PageRank (PPR) sampler, which uses the induced subgraph from the largest PPR scores $\pi$ for a node. According to the paper, both these methods are lightweight and scalable

++ The method is applied to extend GraphSAGE and GAT models and the paper presents empirical results for 5 different datasets. The results are presented in terms of classification performance (F1-score) and computational cost (Inference cost). Benchmarked against 5 baseline models (including a subsampling algorithm), the SHADOW extension gives SOTA performance at a reduced computational cost.

++ The ablation study in the paper, further demonstrates that using an ensemble of subgraphs improves performance and is feasible using the SHADOW framework.

++ The paper is well written and does a good job of putting the work in context and motivating the problem.

Weaknesses:
-- The hyperparameters of the sampling algorithms, while mentioned in the Appendix tables, are not included in the tuning descriptions. I am curious to know if and how these hyperparameter choices affect the performance-cost tradeoff.

-- Is there a performance-cost tradeoff for the subgraph ensemble setting suggested by the paper?

-- Description of the inference cost calculation would be useful.

-- While the paper mentions two other extensions - SHADOW-GCN and SHADOW-GDC, the results do not include them. Is there a reason for that?

Minor comments:
- Labeling Figure 1 with $v$ and $v'$ would make the example much clearer
- The transitions between the theorems and the discussions are sometimes hard to follow. More connections between notation and interpretation would be helpful.
- The “budget” term in the Appendix tables has not been connected to the hyperparameters of the sampling algorithms

---

> ### Author Response · Authors · 2020-11-17
> **Initial Reply to Reviewer 4: Clarification on Conceptual Concerns**
>
> Thanks a lot for the constructive feedback. Indeed, shaDow-GNN is justified from various perspectives both theoretically and empirically. We further highlight the flexibility and generality of our "deep GNN, shallow sampler" principle from the perspectives of
> * GNN architectures: our analysis on expressivity and efficiency applies to various GNNs, ranging from spectral based (e.g., GCN [Theorem 3.2]), spatial based (e.g., GraphSAGE [example of learning $\tau$]), generally injective mapping based (e.g., GIN [revised Theorem 3.3]) to reconstruction based (e.g., PPRGo, GDC [Section 3.3]).
> * Graph samplers: many sampling algorithms fit into our framework. With the k-hop and PPR samplers as examples, one can integrate more samplers based on the "Extensions" paragraph in Section 3.2.
>
> We appreciate your concerns and would address them here and in the revision.
>
> --------------------------------
> ### Re. weakness 1
>
> We tune the sampling hyperparameters based on the following (we will add such descriptions to the revision).
>
> PPR sampler (can be either based on fixed budget $p$ or score thresholding $\theta$, see Section 3.2)
> * If with fixed budget, then we disable thresholding. We range the budget $p$ from 100 to 200 with stride 25.
> * If with thresholding, we set $\theta$ to either 0.01 or 0.05. We still have an upper bound $p$ on the subgraph size. So if there are $q$ nodes in the neighborhood with PPR score larger than $\theta$, the final subgraph size would be $\max\{p, q\}$. Such an upper bound eliminates the corner cases which may cause hardware inefficiency due to very large $q$. We set $p$ to be either 200 or 500 depending on the graph.
>
> k-hop sampler
> * For Table 1 results, we fix $k=2$.
> * Budget $b$ ranges from 5 to 20 with stride 5. In this paper, we always set the same budget for each layer. One may also customize different budgets for different layers as a simple algorithmic extension.
>
> In addition, we agree it would be valuable to study the performance-cost tradeoff by varying the sampling parameters (e.g., $p$ and $b$). We will add the additional empirical results in the next revision.
>
> --------------------
> ### Re. weakness 2
>
> We are conducting additional experiments and will add the results to the next revision.
>
> -------------------
> ### Re. weakness 3
>
> The inference cost in Table 1 is purely a measure of computation complexity, without considering the hardware / implementation factors such as parallelization, batch processing, distributed storage, etc. Consider a $L$-layer GNN generating embedding for a target node $u$. For illustration purposes, here we consider GCN-type of layer aggregation.
>
> For a normal GCN (whether it is *trained* with full-batch or minibatch), the *inference* propagates from all $L$-hop neighbors to all $(L-1)$-hop neighbors, and further to all $(L-2)$-hop neighbors, and so on. Such a process proceeds until the 1-hop neighbors propagate to $u$. Let $n_\ell$ denote the number of $\ell$-hop neighbors. Layer-$\ell$ has $n_\ell$ inputs and $n_{\ell-1}$ outputs. So main layer computation is $X'' = A'\cdot X'\cdot W$, where $A'$ is a $n_{\ell-1}\times n_\ell$ sparse matrix, $X'$ and $X''$ are $n_\ell\times f$ and $n_{\ell-1}\times f$ feature matrices and $W$ is the weight matrix. Suppose the sparsity of $A'$ is $\eta$ (where $\eta$ is approximately $deg/n_{\ell}$), then the cost of this layer is $\eta \cdot n_{\ell} \cdot n_{\ell-1}\cdot f + n_{\ell}\cdot f^2$. Total cost of inference on $u$ is the sum of cost for all the $L$ layers, which can grow exponentially with $L$ due to the explosion of $n_\ell$.
>
> For shaDow-GCN, the main differences from the above analysis are that, $n_{\ell}=n_{\ell-1}=n$ (let $n$ be the subgraph size) and $\eta$ equals the average subgraph degree $d$. The inference cost is simply $L\cdot (n \cdot d \cdot f + n\cdot f^2)$, which is ensured to be linear with $L$.
>
> We will add the formal description of the inference cost computation in the paper. We will also conduct additional experiments to measure the actual inference time on GPUs for these models.
>
> ------------------------------
> ### Re. weakness 4
>
> Initially, we didn't include the results for shaDow-GCN and shaDow-GDC due to page and resource limitations (e.g., limited GPUs). We are performing the additional experiments on these architectures and will include their results in our next revision.
>
> ------------------
> ### Re. minor comments
>
> * We will add the labels in Figure 1. Thanks for the suggestion.
> * We will work on the transition to make the analysis easier to follow.
> * Hyperparameters $b$ and $p$ mentioned in Tables 3 and 4 (Table indices as in the original version) are defined in Section 3.2 -- $b$ is the neighbor expansion factor per extra hop in k-hop sampling. $p$ is the total budget on the number of subgraph nodes in PPR sampling. We will clarify them in Section 3.2.
>
> ----------------
> Thanks again for your valuable comments. We will follow up on this thread to update new results.

---

> > ### Author Response · Authors · 2020-11-23
> > **Summary of Revision 1**
> >
> > We have uploaded our first revision of the paper. In the next revision, we will include the results on GNNs deeper than 5 layers and more results on subgraph ensemble.
> >
> > -------------------------------
> >
> > ### Additional experimental results
> >
> > **[Performance-cost tradeoff]**
> >
> > We have included the accuracy-time tradeoff for the PPR sampler in the current revision. Please check the added Figure 3. In summary, on a pretrained shaDow-GNN, we can flexibility adjust the PPR sample size given the available computation resources, without significant accuracy degradation. In many cases, by simply reducing the PPR sampling size without any retraining, we can achieve 2x to 4x reduction in computation time at the cost of less than 1% accuracy loss.
> >
> > The accuracy-time tradeoff for the case of subgraph ensemble will be included in the next revision.
> >
> > **[Results for other shaDow models]**
> >
> > We have added the results for shaDow-GCN (Table 1), shaDow-GIN (Table 2) and shaDow-JK (Table 2). All the three additional architectures demonstrate the benefits of our shaDow construction by showing significant accuracy gain. As for shaDow-GDC, we would include its results if time permits.
> >
> > ### Additional Technical Contents
> >
> > We have added more details regarding the hyperparameter tuning in Appendix C.
> >
> > We have added the equations for computing inference cost in Appendix B.
> >
> >
> > ------------------------------
> > Please let us know of any concerns.

---

> > > ### Author Response · Authors · 2020-11-24
> > > **Summary of Revision 2: Additional Results on Deep Models and Ensemble**
> > >
> > > In this final revision, we focus on a thorough evaluation of shaDow-GNN with deeper layers. Some additional results on subgraph ensemble have also been included.
> > >
> > > ### Summary of the deep model experiments
> > >
> > > We evaluate normal GNN and shaDow-GNN with depth up to 40. The experiments can be summarized in three aspects:
> > >
> > > * **Understanding non-oversmoothing of shaDow**: to empirically justify Theorem 3.2, we pick the SGC model as the backbone model. SGC is a simplified version of GCN. So by training SGC and shaDow-SGC we can minimize the irrelevant factors such as difficulty in neural network optimization. Results clearly show that shaDow-SGC achieves significantly higher accuracy, especially with more layers.
> > > * **Convergence quality**: We have presented the convergence curves for GCN and shaDow-GCN under 3, 5, 8 and 15 layers. Results show that shallow sampling significantly improves the convergence quality (both the rate and final accuracy). The deeper the model is, the more benefits from the shallow sampler.
> > > * **Test accuracy**: We have presented the test accuracy for models of 3, 5 and 7 layers. Interestingly, on the ogbn-products graph, even though over 98% of the subgraph nodes fall within 2 hops, increasing the model to up to 7 layers can still lead to accuracy improvement.
> > >
> > > ### Summary of subgraph ensemble experiments
> > >
> > > We have included the additional results on subgraph ensemble in Table 3.
> > >
> > > -------------------------------------------
> > >
> > > We would like to thank you again for your thoughtful and thorough review. Your suggestions have helped improve the quality of our work.
> > >
> > > We believe all of the concerns have been addressed.

---

### Official Review · AnonReviewer3 · 2020-11-09
**Interesting paper but some concerns**

**Rating:** 6
**Confidence:** 4

**Review:**

The paper proposes a simple but interesting new graph sampling method for graph neural networks, called “deep GNN, shallow sampler”. Centered on the target nodes, they only sample shallow subgraphs within $L_0$-hop neighborhood and then run an $L$-layer GNN ($L>L_0$) on these subgraphs and aggregate their embeddings.  In this way, they can limit the message passing only within a shallow neighborhood to exclude noisy nodes; and they can also improve the expressivity by using deep GNN. To my understanding, the two most similar works are GraphSAGE and GraphSAINT. Compared to GraphSAGE, it samples the subgraphs instead of just $l$-hop nodes (it means they may contain more edges/circles), and it can be more expressive; compared to GraphSAINT, it requires the samples to be centered around target nodes and shallow, and it also changes the way of subgraph ensemble and makes it applied to the testing phase.
In general, I think this paper has concentrated contributions and it could be impactful in practical use, but I still have some concern.
1.	When the authors compare the expressivity with “shallow GNN, shallow sampler”, it seems to me that $L=L_0+1$ is enough (we cannot have edges with distance more than $L_0+1$ to the target node). Adding more layers of GNN does not help with the expressivity. Even if my assumption is false and adding more layers is helpful, the authors also need to do more sensitive analysis on the impact of different numbers of GNN layers.
2.	Experiments: 3-layer or 5-layer GNN can be regarded deep, we know directly running the original model does not perform well due to the over-smoothing issue. However, there are many tricks to solve it, such as Jumpingknowledge, Skip Connection, DropEdge. I think the baseline models should use one of these tricks to make the accuracy comparison more reasonable, and it will also be interesting to see whether these tricks are necessary for the proposed model.
3.	One of the main advantages of graph sampling is its efficiency. In the experiments, the authors show the computation cost of each target node. This is OK but it seems to me that showing the running time is a more intuitive and better way. Also, it is better to add the comparison with GraphSAINT in terms of running time or epochs (e.g. adding the GraphSAINT results in Figure 2).

---

> ### Author Response · Authors · 2020-11-17
> **Initial Reply to Reviewer 3: Clarification on Conceptual Concerns**
>
> Thank you very much for the constructive feedback. You are right that compared with a wide class of GNNs, the proposed shaDow-GNN improves:
> * expressivity -- whether the aggregation is spectral based (e.g., GCN [Theorem 3.2]), spatial based (e.g., GraphSAGE [learning the $\tau$ function]), generally injective mapping based (e.g., GIN [revised Theorem 3.3]) or graph reconstruction based (e.g., PPRGo, GDC [Section 3.3]).
> * efficiency -- since our method fills the gap that there lacks a scalable minibatch algorithm for the inference phase (as noted by the reviewer and also in the paper, GraphSAINT minibatching only applies to training).
>
> Your concerns are all very reasonable. We will address them in this discussion thread and then update the paper correspondingly.
>
> ---------------------------------------------
> ### For your first concern
>
> Theoretically, on the $L$-hop subgraph, an $L'$-layer GNN can still be quite useful when $L' > L+1$:
>
>
> (Note, the following part between the && markers is overlapping with the reply to Reviewer 2)
>
> &&
>
> **[Case 1]**: Suppose we use a $L'$-layer GraphSAGE to approximate a function $\tau$ on the subgraph $G_{[u]}^L$. We consider an example $\tau$ as computing the unweighted mean of the subgraph node features. In this case, the error of approximating $\tau$ converges to zero when $L'$ goes to infinity. The error can still be significant if $L'$ is not sufficiently larger than $L$. The desired depth $L'$ is determined by the mixing time of the subgraph adjacency matrix. See the discussion on the second example regarding Figure 1.
>
> **[Case 2]**: Consider a more "powerful" GNN when $f_1$ and $f_2$ of Equation 3 perform injective mapping. Recall the following two facts:
> * On the full graph, such a GNN of $L_0$ layers is as discriminative as the 1-WL test running $L_0$ iterations.
> * 1-WL may take up to $O(N)$ iterations to converge where $N$ is the full graph size.
>
> Now extend to the shaDow case. Consider two target nodes $u$ and $v$ with their corresponding $L$-hop subgraphs $G_{[u]}^L$ and $G_{[v]}^L$. An $L'$-layer GNN will output different embeddings for $u$ and $v$, if $u$'s color assigned by $L'$-iteration 1-WL on $G_{[u]}^L$ is different from $v$’s color assigned by $L'$-iteration 1-WL on $G_{[v]}^L$. The coloring of $u$ and $v$ can keep changing for $O(N)$ iterations (here $N$ is the size of $G_{[u]}^L$ and $G_{[v]}^L$). Thus, increasing the GNN depth on $L$-hop subgraphs can improve discriminative power for up to $O(N)$ layers. Empirically, the subgraph size $N$ is set between 100 to 200.
>
> So from the above theoretical perspective, on a shallow subgraph, we may see the benefit of deepening the GNN for a truly large depth. In Theorem 3.2, since the statement only specifies a general condition of $L' > L$, it suffices to discuss just the $L'=L+1$ case in the proof. On the other hand, we agree with you that discussing the benefit beyond $L'=L+1$ would clarify the understanding. Therefore, we will add the above analysis to the paper.
>
> &&
>
> From the practical perspective, a deep GNN may not always lead to accuracy gain due to difficulties in optimization and generalization. We are currently conducting additional experiments on different numbers of GNN layers. We will include the results in the next revision.
>
> -------------------------------------------------------
> ### For your other two concerns
>
> We will include the additional experimental results in the next revision. We will follow up on this thread to for further detailed discussions.
>
> Thanks again for your time and consideration.

---

> > ### Author Response · Authors · 2020-11-23
> > **Summary of Revision 1**
> >
> > We have uploaded our first revision of the paper. In the next revision, we will include the results on GNNs deeper than 5 layers and more results on subgraph ensemble.
> >
> > ---------------------------------
> >
> > ### Additional experimental results
> >
> > **[Other methods for improving deep GNN performance]**
> >
> > We have rerun all experiments in Table 1 (for both the baseline and shaDow models) by tuning the additional DropEdge parameters. We observe that adding DropEdge leads to slightly better accuracy. DropEdge benefits the baselines due to the alleviation of oversmoothing, and benefits shaDow-GNNs due to its regularization effects.
> >
> > We have also included the results for shaDow-JK (Table 2), shaDow-GIN (Table 2) and shaDow-GCN (Table 1). In particular, we find that for the normal GNN setup, the effect of skip connection by JK-Net is similar to that of DropEdge. Furthermore, shaDow-JK achieves significantly higher accuracy than the normal JK-Net.
> >
> > **[More evaluation on the sampling cost]**
> >
> > We have included the execution time of samplers on real machines. In the added Figure 2, the execution time of PPR sampling is less than the GNN execution time on GPU. Furthermore, we show the accuracy-time tradeoff of various sampling budgets in the added Figure 3.
> >
> > **[GraphSAINT training time in the convergence curve]**
> >
> > We will add to the (original) Figure 2 the GraphSAINT convergence curve w.r.t. training time in the next revision. Such addition will be presented together with other new results on subgraph ensemble.
> >
> > **[Sensitivity analysis on number of layers]**
> >
> > We will add this in the next revision. We have observed empirical evidence that shaDow-GCN does not oversmooth.
> >
> > ### Other modifications
> >
> > We have updated Theorem 3.3 and its proof based on the review and our previous reply.
> >
> > ------------------------
> >
> > Please let us know of any concerns regarding the current revision.

---

> > > ### Author Response · Authors · 2020-11-24
> > > **Summary of Revision 2: Additional Experimental Results**
> > >
> > > We have uploaded our final revision. The main changes compared with the previous revision are as follows:
> > >
> > > ### Evaluation on deeper models
> > >
> > > We have thoroughly evaluated how the increase of model depth would impact the shaDow-GNN accuracy, from the following perspectives:
> > > * We have presented the convergence curves for GCN (GraphSAINT trainer) and shaDow-GCN under 3, 5, 8 and 15 layers. Results show that shallow sampling significantly improves the convergence quality (both the rate and final accuracy) for the 15-layer model.
> > > * We have presented the test accuracy for models of 3, 5 and 7 layers. Interestingly, on the ogbn-products graph, even though over 98% of the subgraph nodes fall within 2 hops, increasing the model to up to 7 layers can still lead to accuracy improvement.
> > > * We have further implemented shaDow-SGC and compared with the normal SGC. The SGC model performs the same aggregation operation as GCN and so we think it as appropriate to validate Theorem 3.2. We increase the power on the adjacency matrix from 1 to 40, and results clearly show that shaDow-SGC achieves significantly higher accuracy, especially with higher power (corresponding to deeper layers).
> > >
> > > ### Evaluation on subgraph ensemble
> > >
> > > We have included more results to demonstrate the benefit of ensemble. In the added Table 3, ensemble of PPR and 1-hop improves test accuracy compared with PPR alone or 1-hop alone.
> > >
> > > The comparison with GraphSAINT is included in the added Figure 4, as part of the analysis on deeper models.
> > >
> > > -----------------------------------
> > >
> > > As the rebuttal period is ending soon, we thank you again for your valuable feedback. We believe we have addressed all of your concerns in the revision.

---

### Author Response · Authors · 2020-11-24
**End of Rebuttal: Thanks**

As the rebuttal period is ending soon, we would like to thank again the four reviewers for their valuable feedbacks.

We believe we have addressed every concern in our final revision.

Thank you!

---

### Author Response · Authors · 2021-10-05
**Paper Accepted to NeurIPS 2021**

Revision of our paper has been accepted to NeurIPS 2021, with the new title "**Decoupling the Depth and Scope of Graph Neural Networks**".

Code release: https://github.com/facebookresearch/shaDow_GNN

---

### Decision · Program_Chairs · 2021-01-07
**Final Decision**

**Decision:**

Reject

**Comment:**

In this paper, the authors propose a simple yet interesting new graph sampling method for graph neural networks.  It addresses the two main problems that GNNs have not previously been extended to deep GNNs: expressivity and computational cost.  Through experiments, the authors show the effectiveness of the proposed algorithm.

Overall, the proposed approach is interesting.  However, the reviewers were still not convinced by the response, and the paper is still below the acceptance threshold.  I encourage the authors to revise the paper based on the reviewer's comments and resubmit it to a future venue.